# Single nucleus transcriptomics of ventral midbrain identifies glial activation associated with chronic opioid use disorder

Julong Wei[1], Tova Y. Lambert [2], Aditi Valada[2], Nikhil Patel[3], Kellie Walker[3], Jayna Lenders[3], Carl J. Schmidt[4], Marina Iskhakova[2], Adnan Alazizi[1], Henriette Mair-Meijers[1], Deborah C. Mash[5,8], Francesca Luca[1,6,7,8], Roger Pique-Regi [1,6,8], Michael J. Bannon [3,8] & Schahram Akbarian [2,8] ✉

Dynamic interactions of neurons and glia in the ventral midbrain mediate reward and addiction behavior. We studied gene expression in 212,713 ventral midbrain single nuclei from 95 individuals with history of opioid misuse, and individuals without drug exposure. Chronic exposure to opioids was not associated with change in proportions of glial and neuronal subtypes, however glial transcriptomes were broadly altered, involving 9.5 – 6.2% of expressed genes within microglia, oligodendrocytes, and astrocytes. Genes associated with activation of the immune response including interferon, NFkB signaling, and cell motility pathways were upregulated, contrasting with down-regulated expression of synaptic signaling and plasticity genes in ventral midbrain non-dopaminergic neurons. Ventral midbrain transcriptomic reprogramming in the context of chronic opioid exposure included 325 genes that previous genome-wide studies had linked to risk of substance use traits in the broader population, thereby pointing to heritable risk architectures in the genomic organization of the brain's reward circuitry.

Ventral midbrain (VM), including the ventral tegmental area (VTA) and substantia nigra (SN)[1,2], is important for mediating habitual behaviors and salience of cues associated with drug use, as well as withdrawal-related anhedonia and dysphoria[3,4]. It has become increasingly clear in recent years that, in addition to the well-established roles of DA and non-DA (e.g., GABAergic) neurons, the VM's glial and other non-neuronal populations may play an important role for drug responsiveness and substance use. To mention just three representative examples, excessive activation of VM microglia is thought to disrupt chloride homeostasis in GABA neurons, which in turn, negatively affects opioid and stimulant-induced dopamine release and associated reward behaviors[5]. Likewise, VM astrocytes play an essential role in drug-induced synaptic plasticity in DA neurons[6], a reflection of astrocytic regulation of neuronal glutamine supply and glutamatergic neurotransmission[7]. Finally, oligodendrogenesis in VM is essential for morphine-mediated reward behavior, and proliferation and differentiation of VM oligodendrocytes (ODCs) is regulated by the firing activity of their surrounding dopaminergic neurons[8].

However, despite these intriguing mechanistic studies in animal models, the functional and clinical significance of VM glial populations

[1]Center for Molecular Medicine and Genetics, Wayne State University School of Medicine, Detroit, MI 48201, USA. [2]Department of Psychiatry, Department of Neuroscience and Department of Genetics and Genomic Sciences, Friedman Brain Institute Icahn School of Medicine at Mount Sinai, New York, NY 10029, USA. [3]Department of Pharmacology, Wayne State University School of Medicine, Detroit, MI 48201, USA. [4]Department of Pathology, University of Michigan School of Medicine, Ann Arbor, MI 48109, USA. [5]Department of Neurology, Miller School of Medicine, University of Miami, Miami, FL 33136, USA. [6]Department of Obstetrics and Gynecology, Wayne State University, Detroit, MI 48201, USA. [7]Department of Biology, University of Tor Vergata, Rome 00133, Italy. [8]These authors jointly supervised this work: Deborah C. Mash, Francesca Luca, Roger Pique-Regi, Michael J. Bannon, Schahram Akbarian. ✉e-mail: schahram.akbarian@mssm.edu

in subjects diagnosed with substance use disorder remains unexplored. To this end, cell-specific transcriptomic profiling of VM dissected from human post-mortem brain could deliver critical insights. This task is particularly urgent for opioid use disorder (OUD), considering that opioid overdose (OD) is now the leading cause of accidental deaths in the United States, with ~70,000 deaths annually reflecting a >8-fold increase over the course of just two decades[9]. However, to date, with the exception of a single study profiling RNA from VM bulk tissue in a limited cohort of opioid users and controls[10], no knowledge exists about genome-scale dysregulation associated with chronic opioid exposure and overdose. Further, RNA-seq profiling of VM bulk tissue is insufficient to disentangle cell type-specific contributions in neuropsychiatric disease[11].

Of note, in recent pilot studies exploring adult postmortem human VM single cell genomics (by 10x chromium single nuclei transcriptomic profiling; Smajic and colleagues[12], and others[13], reported very high recovery rates of glial and other non-neuronal nuclei in these 'dopaminergic' brain structures, with >95–96% of the total population of nuclei recovered from the SN contributed by prototypical glia, including ODCs and their precursors, astrocytes, and microglia. In contrast, DA and GABA neurons taken together contributed only a very minor (< 4%) share of nuclei in this type of single nuclei RNA-seq assay[12,13]. This approach thus lends itself to an in-depth characterization of opioid-related changes in gene expression in these relatively under-studied glial cell types in human VM.

Here, we present our findings from a transcriptomic study at single nuclei resolution in the VM, built from two independent opioid misuse-control cohorts (totaling 95 subjects) from different geographical areas in the U.S. We report reproducible alterations affecting hundreds of microglia-, astrocyte- and oligodendroglia-associated transcripts. Of note, these widespread, cell type-specific disruptions of the glial VM transcriptome in individuals who died by opioid overdose occurred in the context of completely conserved cellular composition, with stoichiometric proportions for all neuronal and glial subtypes indistinguishable between the disease and the control group. Our findings point to alterations of gene expression in individuals who died by opioid overdose, indicative of neuroinflammation and activation of cytokine signaling in the VM, primarily affecting microglia and astrocytes, with additional alterations of oligodendrocyte-specific transcriptomes. More broadly, the dataset presented here will provide a human neurogenomics resource at single cell resolution for the wider field of drug abuse research.

## Results

### Chronic opioid exposure does not alter the cellular composition of the ventral midbrain

We generated VM single nuclei RNA-seq libraries for 95 brain donors (84M/11F), including 45 subjects with documented histories of opioid abuse and overdose, and 50 demographically-matched, opioid-free control subjects, collected from two geographically distinct regions within the U.S. (greater Detroit area, Michigan and Miami, Florida) (Fig. 1A, Table S1, Data S1). Each VM sample included both substantia nigra and the adjacent ventral tegmental area (SN/VTA) (see Figs. S1, S10A and Methods). Nuclei were processed in pools of 3–4 brains of diseased mixed with control brains, using the 10X Chromium system followed by Illumina sequencing, read alignment and processing by 10X Cellranger. Each single nucleus was matched to a specific donor using Demuxlet, confirming a 100% match by donor by pool against the background of all 95 donors/95 samples (1 sample/donor) (Fig. S1A). After removal of doublets and quality-control filtering (see Methods), we obtained a total of 212,713 transcriptionally profiled single nuclei, each unique to a singular donor (median, 2008 nuclei/donor). We collected 2696–21,363 (median, 8274) reads/nucleus (Data S2) and measured the expression of 1383–5079 (median, 3070) genes/nucleus. Total numbers of single nuclei/specimen, genes called/single

nucleus/specimen and read depth/nucleus/specimen showed no significant differences between VM of individuals who died by overdose and controls (Fig. S1B–D).

Resolving the entire collection of 212,713 nuclei by cluster analysis in Seurat v.4.0 with 2000 highly variable genes and 50 harmony-adjusted principal components (PC) produced in the Uniform Manifold Approximation and Projection (UMAP) plot 10 principal cell types, further confirmed by computational annotation to a reference dataset built from 18 SN samples from an independent study[14] (Fig. S1E–G) and by marker gene expression (Fig. 1b, c). Representative examples of gene expression uniquely defining a specific cell type include oligodendrocyte transcription factors 1 & 2 (OLIG1/2) for oligodendrocyte precursor cells (OPCs), myelin-associated oligodendrocyte basic protein (MOBP) and myelin basic protein (MBP) for ODCs, the classical astrocytic markers Aquaporin-4 (AQP4) and glial fibrillary acidic protein (GFAP), complement and chemokine signaling genes C3 and CX3CR1 for microglia, and various markers specific to each of the remaining cell types including endothelium, pericytes, ependyma and T lymphocytes (Fig. 1b, Fig. S2A–C). Furthermore, as expected for VM, the neuronal subpopulation split into dopaminergic (DA) and non-dopaminergic (Non-DA) (Fig. 1b), with the former showing expression for dopamine biosynthetic genes including dopa decarboxylase and tyrosine hydroxylase (DDC, TH) and the latter separating into a larger subgroup of gabaergic neurons defined by expression of GABA biosynthetic enzymes glutamic acid decarboxylase GAD1, GAD2 and vesicular GABA transporter VGAT SLC32A1, and a smaller subgroup of glutamatergic neurons expressing vesicular glutamate transporters VGLUT1/2 (SLC17A6/7). Furthermore, in line with previous studies with single cell resolution in rodent VTA/SN[15,16], our samples showed considerable heterogeneity for some of the established markers for DA neuron subtyping, including ALDH1A1, SOX6, and SLC17A6 (Fig. S2D, E).

We then asked whether opioid exposure altered the proportions of various cell types, including of the various glial populations that were the focus of the present study. In controls, ODC and their precursors (OPC) taken together comprised 64.3% of all VM nuclei, a proportion that is highly consistent with an independent dataset[14], followed by astrocytes (15.0%) and microglia (13.5%). In contrast, DA and non-DA (including GABA) neurons together accounted for 3.8% of VM nuclei, while pericytes, endothelium, T-cells, and ependyma together represented the remaining 3.1% of nuclei in our VM specimens from control individuals (Fig. 1d). Of note, our VM specimens from subjects who died by overdose showed very similar numbers and proportions for each cell type compared to controls. We conclude that chronic opioid exposure and overdose is not associated with proportional shifts among the neuronal and glial constituents in the VM (Fig. 1d). Consistent with this observation, disease and control groups from each of our two collection areas, when plotted separately into our UMAP coordinates, showed highly similar distributions by cell type (Fig. 1e).

### Hundreds of glial transcripts show altered expression in opioid-exposed midbrain

Of note, transcripts for each of the four G-protein coupled opioid and opioid-related receptors, including the 'classical 3' OPRM1 (mu) and OPRD1 (delta) and OPRK1 (kappa), plus OPRL1 (nociceptin), were readily detectable among the various glial cell types in the VM, with particularly robust expression of OPRM1 in the microglia (Fig. S3). These findings, which are consistent with previous reports on cell-specific expression, and functional, ligand-binding and mutant-mice studies[17,18] would suggest that opioid exposure could have direct effects on VM glia in addition to adaptations mediated by drug-related neuronal signaling changes. To further explore this scenario, we computed cell-type specific differential gene expression (DEG) by diagnosis (history of opioid use and overdose vs. drug-free control).

Focusing on autosomal gene expression, we first obtained a counts matrix of 30,801 genes, presenting in 531 combinations (each comprised of a minimum of 30 single nuclei) defined by sample and cell type. Importantly, the Detroit and Miami cohorts, each analyzed separately, showed on a genome-wide scale strong positive correlations for cell-type specific transcriptome differences between their respective disease and control brains, speaking to the generalizability of findings. Specifically, the highest z-score correlations (Miami cohort vs. Detroit cohort) were observed for OPC ($R = 0.44$), ODC ($R = 0.32$), astrocytes ($R = 0.30$) and microglia ($R = 0.29$) ($P < 2.2^{-10(-16)}$) (Fig. S4A). We therefore conducted DEG analysis by combining the Detroit and Miami cohorts, using sex, genetic ancestry (genotype PCs), age, and postmortem confounders (e.g., brain pH) as covariates. Our initial round of covariate-corrected DEG analysis, with a log fold-change threshold >0.25 and False-Discovery Rate (FDR) corrected $P < 0.1$, identified 5239 DEGs from a total of 25,728 genes included in the DEG

analyses, with 2999 up- and 2381 down-regulated, with many of these genes dysregulated in more than one cell type (Data S3). However, the impact of opioid OD on the genome-wide VM transcriptome showed striking disparities by cell-type due to the preponderance of glial-specific alterations. Thus, 9.5% (2131/22,536) of microglia-, and 7.5–6.2% (1503/20,050 to 1462/22,536) of astrocyte- and ODC/OPC-associated transcripts were differentially regulated in comparison to drug-free control subjects. In sharp contrast, only 0.32% (70/21,881) of the GABA/non-DA neuronal transcriptome, and none (0/15,041) of DA neuron expressed transcripts, were detected as altered in individuals who died by OD (Data S3). Furthermore, while the largest share of DEG, or 2131/5239 (42.5%) was contributed by the microglial population, each of the major glial subtypes shared between 246 and 395 DEGs with at least one additional glial subtype (Fig. 2), with shared directionality in a large majority of these DEGs (83–95%, depending on cell type) (Fig. S4B).

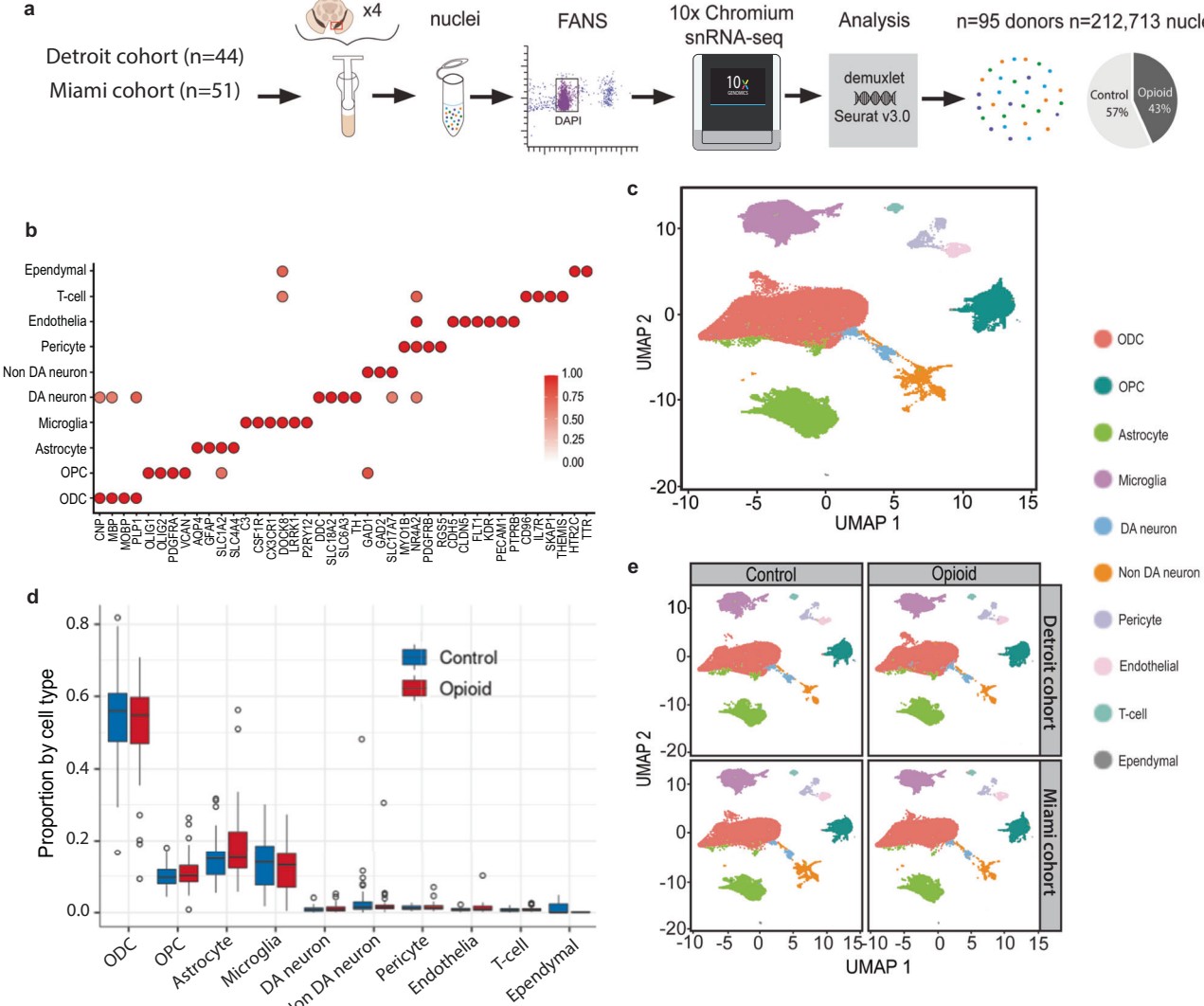

**Fig. 1 | Consistent proportions of VM cell types across specimen collection sites and cohorts. a** Experimental design and workflow for $N = 95$ VM samples, including collection at two different geographical areas, pooling of 3–4 VM specimens, nuclei purification by FACS, and 10x chromium snRNA-seq pipeline and genetic demultiplexing yielding a total of 212,713 nuclei. **b** Marker gene expression for each of the 10 glial and neuronal subpopulations as indicated. DA, dopaminergic neuron; non-DA, non-dopaminergic neurons; ODC, oligodendrocyte; OPC, oligodendrocyte precursor cell. Color represents the ratio of average gene expression across cells in the cell type relative to maximum in the most highly enriched cell type. **c** Uniform Manifold Approximation and Projection (UMAP) plot showing the identified 10 major cell types by cluster, as indicated, for total collection of $n = 212,713$ nuclei. **d** Box-and-whisker representation of the proportion of the 10 major cell types in each individual (box represents the first quartile, the median, and third quartile, while the whisker spans the 1.5x interquartile range of the first and third quartile); split by diagnosis as indicated red, opioid-related death and blue, control. **e** VM cell type composition by UMAP plot, shown for diseased and control individuals separately for each of the two collection sites. **a–e** $N = 95$ samples, Source data are provided as a Source Data file.

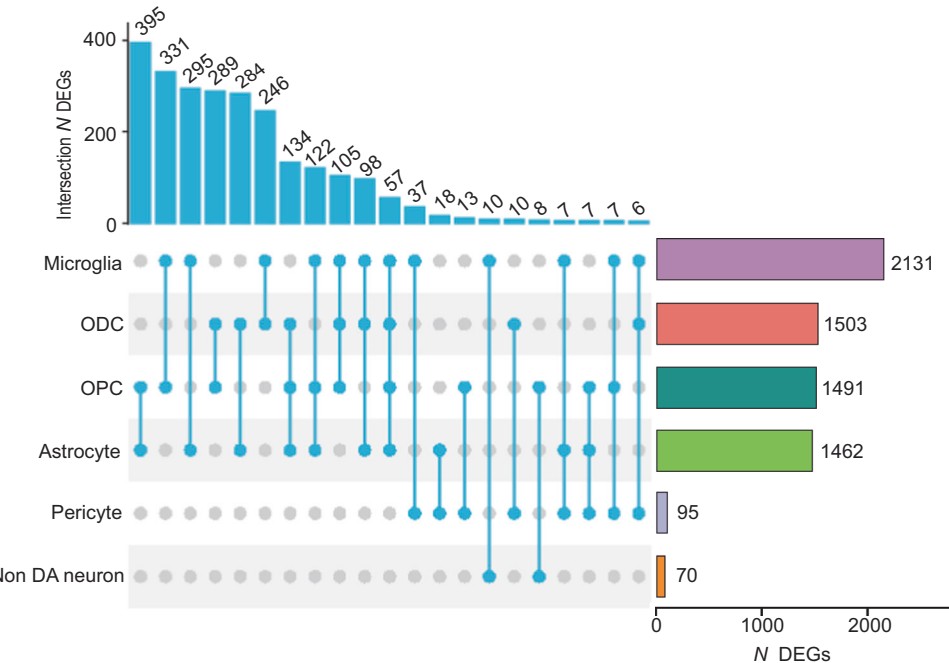

**Fig. 2 | Representation of differentially expressed genes (DEGs) shared across VM cell types.** (left) Counts of shared DEGs across cell types, as indicated by the UpSet diagram. (right) Number ($N$) of DEGs for each cell type, from (top) microglia, $N = 2131$ DEGs to (bottom) non-dopaminergic neurons, $N = 70$ DEGs. $N = 95$ samples, Source data are provided as a Source Data file.

Unbiased subclustering of glial cell types revealed at least two subpopulations of oligodendrocyte, and of microglia nuclei, each of which were broadly represented in the majority of the diseased and control individuals (Fig. S5A, Data S4). For example, ODC subcluster '0' was defined by elevated expression of *OPALIN* and other marker genes that define myelin-forming ODCs, while ODC subcluster '1' showed much higher expression of *S100B, RBFOX1* and various other marker genes previously linked to mature (aged) and stressed ODC[12,19] (Fig. S5A, B, Data S4). However, correlation matrices summarizing DEG between diseased individuals and controls across all glial and neuronal subpopulations (Fig. S5C), in addition to subtype specific DEG analysis and proportional counts of nuclei in diseased and control brains (Fig. S5D), confirm that oligodendrocyte-specific transcriptional alterations are not limited to the subpopulation of aged and stressed nuclei while also affecting younger, myelin-forming ODC. Likewise, correlational analyses confirmed that microglia-specific alterations in individuals who died by overdose affect multiple types of microglia (Fig. S5C). This included subtype '0', or a large group of microglial nuclei defined by higher expression of interleukin *IL18* and heat shock protein *HSPB1*, two molecules that reportedly promote neuroinflammation in adult human brain[20–22], and higher expression of additional regulators of cytokine signaling such as SOCS6 which is thought to affect the interaction between microglia and midbrain dopaminergic neurons[23] (Data S4, Fig. S5A).

**Transcriptomic signatures of opioid-exposed midbrain include glial activation and downregulation of synaptic functions in non-dopaminergic neurons**

We noted that expression of molecules broadly linked to glial activation and neuroinflammation, including STAT3, STAT5A/B and other members of the Signal Transducer and Activator of Transcription (STAT) transcription factor family[24,25] were upregulated in various glial populations of OD VM (Data S3). Therefore, to explore this phenomenon on a genome-wide scale, we next conducted cell-type specific gene ontology (GO) over-representation analyses. Indeed, up-regulation of immune response pathways including, for example,

interferon, NFkB signaling, and cell motility ranked top in all glial populations of VM, including astrocytes, pericytes, microglia, and ODC/OPC (Figs. 3, S6, S7, Data S5). In striking contrast to these types of glial activation, top ranking GOs enriched in neuronal DEGs revealed downregulation of functions related to synaptic connectivity including ionotropic glutamate receptor signaling, long-term potentiation, neurite extension and others, in conjunction with increased chromatin repression by histone (H3- lysine 9) methylation (Figs. 3, S7, Data S5). These decreases in synaptic gene expression were highly specific to neurons and not observed in glia.

Furthermore, down-regulated expression in ODC from opioid-exposed VM included multiple GO-defined mitotic spindle genes (Fig. 3, Data S5), such as *AURKA, FIGNL1*, and *KIF11*. Importantly, the function of these genes extends beyond mitosis as they maintain expression in interphase nuclei to regulate microtubular structures[26–28] and, in case of *TMEM67*, are linked to white matter tract alterations in the human midbrain[29]. Therefore, altered expression of these genes could indicate potential cytoskeletal alterations in differentiated, postmitotic ODC from individuals who died by overdose.

Having shown widespread reprogramming of the nuclear transcriptome in multiple cell populations of VM from individuals who died by opioid overdose, with activation of immune signaling in multiple glial populations and decreased synapse related gene expression in (non-dopaminergic) neurons, we then asked whether these observations would be broadly reproducible by gene expression profiling from whole cells or even bulk tissue. To this end, we compared the VM cell type-specific DEG between opioid users and control subjects in the current study for each VM cell type to an earlier, smaller ($N = 50$) study[10] involving RNA-seq profiling of bulk VM tissue (NB: there was no overlap in brain donors used in the two studies). We observed positive z-score correlations that were strongest for the astrocytic ($R = 0.19$), ODC ($R = 0.18$) and microglia ($R = 0.17$) populations (Fig. S8). These correlations were highly significant ($p < 1.96 \times 10^{-4}$). In addition, previously reported GO enrichments from bulk VM tissue analysis strikingly resonated with the findings presented here, including upregulation of NFkB signaling and inflammatory, cytokine and

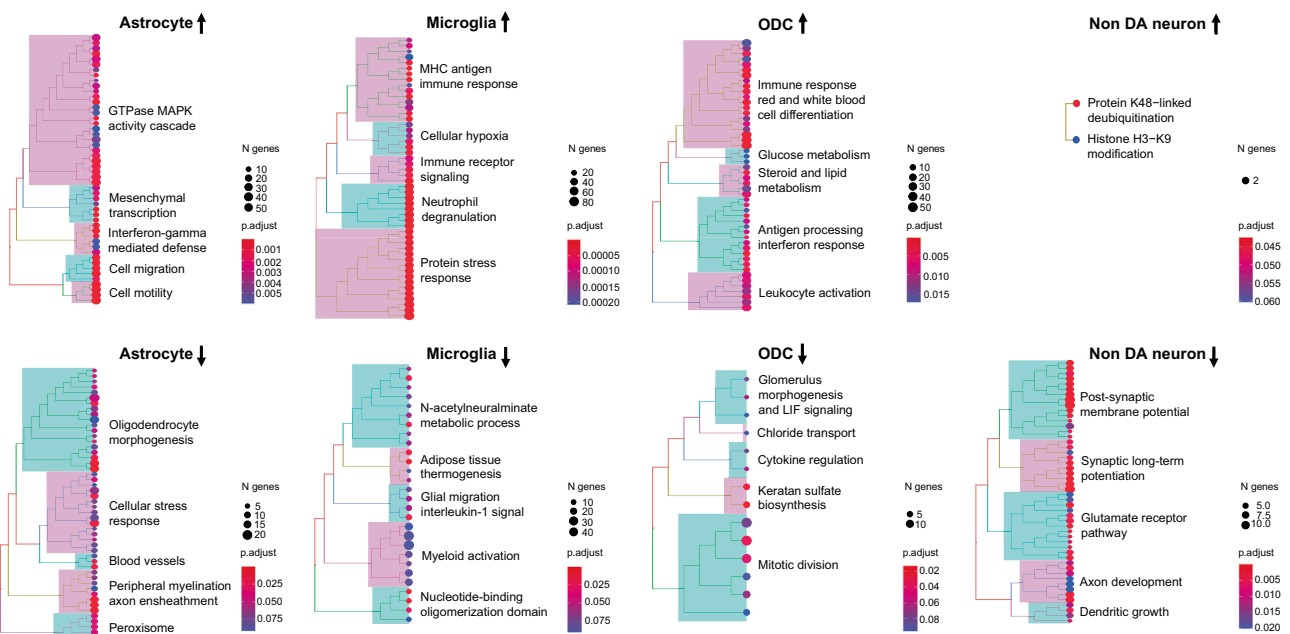

**Fig. 3 | Pathway enrichments of up- and down-regulated VM transcripts by cell type.** Biological Process GO enrichments as indicated. *P*-values, one-sided Fisher's exact test with Benjamini–Hochberg FDR correction (x12, 6 cell types and 2 directions). For additional details, see Figs. S6, S7 and Data S5, including gene ratios (Data S5, column E). *N* = 95 samples, Source data are provided as a Source Data file.

hypoxia response pathways, and were highly reminiscent of the glial activation patterns identified in the present study. Many individual glial transcripts significantly altered in our opioid-exposed VM specimens in a cell-specific manner replicated the most robust changes seen in the aforementioned VM whole-tissue RNA-seq study[10] (Fig. 4). For example, among the group of immune signaling genes with increased expression in opioid-exposed VM, up-regulated Interleukin 4 receptor (*IL4R*) expression in VM bulk tissue was previously reported to be strongly predictive of diagnostic categorization (opioid vs control)[10]. According to the cell-specific results presented here, the *IL4R* gene is highly expressed in microglia, consistent with a role in anti-inflammatory reprogramming of microglia and macrophages after brain or nerve injury[30]. Another top-ranking gene in the VM opioid overdose whole-tissue RNA-seq study was MAP3K6 kinase, a gene predominantly expressed by astrocytes and implicated in angiogenesis[31]. In the current study, MAP3K6 and the related molecule MAP3K7, implicated in abnormal neurovascular regulation in OUD and overdose brain[32], were confirmed as being dysregulated specifically within astrocytic nuclei. Furthermore, notable pathway alterations in the OPC/ODC cell population from the diseased individuals of the present study included CNS injury-mediated differentiation programs, including the bZIP MAF transcription factor MAFF[33] and the cell cycle regulator Cyclin Dependent Kinase Inhibitor *CDKN1A* which, again, were among the top scoring DEG in the previous VM bulk tissue-based gene expression study (Fig. 4). Furthermore, the latter gene is robustly induced in the ventral striatum of morphine-exposed mice[34] and in VM of subjects diagnosed with cocaine use disorder[35], implicating a broader role for *CDKN1A* in addiction biology beyond opioids in the midbrain.

We note that in both the previous VM bulk tissue RNA-seq[10] and the present VM single nuclei RNA-seq studies, neuron-specific transcriptional programs for structural and functional neuronal connectivity, and synaptic transmission were downregulated in the opioid group. According to the present study, this effect is driven by transcriptomic alterations in the non-dopaminergic neurons. Furthermore, the AP-1 transcription factor and early response gene, FOSL2, previously found to be induced in rodent addiction circuitry by chronic

morphine administration[36], showed increased expression in numerous neuronal and glial cell populations of our sn-RNA-seq study, including OPC, microglia and non-dopaminergic neurons. Similarly, FOSL2 was among the top scoring DEGs in the bulk VM RNA-seq study (Fig. 4).

To summarize, the current study identified robust changes in VM gene expression and associated GO pathways in individuals with a history of opioid use and OD that are consonant with a previous report but reveal the cell type-specific nature of these changes, which can be characterized as broad glial activation with a prominent representation of immune signaling pathways, in conjunction with down-regulation of glial support and neuronal signaling genes. This transcriptomic signature observed was independent of the period of specimen collection or geographical location of the cohorts.

## Cell-specific DEGs associated with genetic risk for addiction disorder

Next, we wanted to explore whether any cell type specific DEGs in our opioid exposed disease cohort are associated with the genetic risk for addiction disorder. Of note, while OUD is considered moderately heritable, with an estimated 60% of population variability attributable to genetic factors[37,38], to date only two or three loci have been genome-wide reproducibly linked to opioid use and substance-associated traits[39]. Of note, these loci include the cell motility regulator, *SCAI* (chr. 9q33.3)[39], which in our study is significantly downregulated in VM microglia from individuals who died by opioid overdose (Data S3). However, opioid exposure is broadly associated with genetic risk for substance use and dependence overall. Therefore, we wanted to explore whether any gene expression alterations in our diseased individuals, including cell type-specific dysregulation, could match a broader list of genes linked to heritable substance use traits. To this end, we screened PhenomeXcan[40], a resource for transcriptome-wide association studies linking genes to phenotypes by genetically predicted variation in gene expression. We focused on brain gene expression and population-scale substance use phenotypes in PhenomeXcan including 40 traits related to caffeine, nicotine, alcohol, and marijuana consumption or dependence[41] (Data S6) and, for comparison, a number of medical traits associated with hundreds or

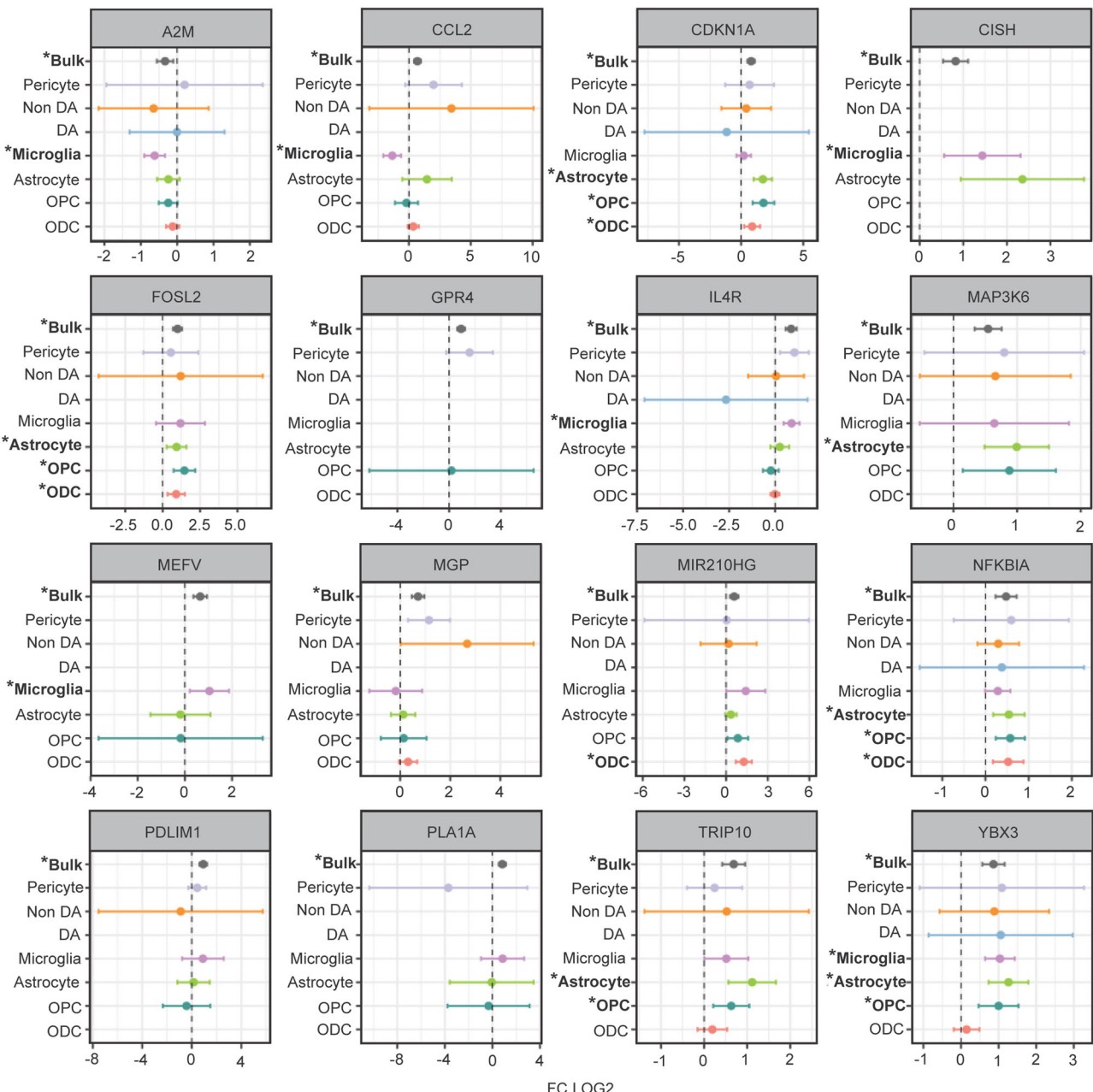

**Fig. 4 | Differentially expressed genes in VM from opioid users: Comparison of whole-tissue and cell-specific data.** Forest plots of selected DEGs (FDR P < 0.1) in opioid overdose VM, comparing (x axis log fold change compared to control) previous VM bulk tissue RNA-seq[10] to cell-type specific profilings in the current study. Bars represent 95% confidence interval (LFC + /−1.96 S.E.). *bold font marks P value from DESeq2 (Wald test following chi-squared distribution with 1 degree of freedom), FDR (Benjamini−Hochberg) P < 0.1. See also Fig. S8. N = 95 samples (brain donors). Source data are provided as a Source Data file.

thousands of significant genes in the PhenomeXcan resource. We found no specific enrichment of our cell type-specific DEG datasets with traits linked to any specific drug or substance use, or to a general addiction risk score as modeled by Hatoum and colleagues[42] in their recent TWAS studies using GTEX and PsychENCODE datasets as input (Fig. S9A, Data S7, S8). However, among the 1260 PhenomeXcan genes linked to a substance use trait and expressed by at least one cell type in the present study, 325, or 25.8%, were called as significantly altered in our study, including 149 DEGs in microglial nuclei, and 100–70 DEGs in astrocytes and ODCs and their precursors, respectively, and only very minimal contributions from some of the remaining cell types including pericytes and non-dopaminergic neurons (Fig. S9B, Data S7). This included 17 genes called as DEG in one or more VM cell types of the present study, and in the VM bulk tissue RNA-seq study[10], with 14/17 of

genes showing the same direction of change across studies (Data S7). Among these, *NUPR1*, a regulator of chromatin acetylation, showed upregulation of expression in VM tissue[10] and VM astrocytes of our disease cohort, and was identified as a risk gene in caffeine, alcohol and marijuana abuse and dependence Interestingly, NUPR1, also known as *STRESS PROTEIN 8* or p8, sensitizes astrocytes to oxidative stress when upregulated[43], in effect exerting a protective effect by decreasing the production of oxygen radicals[44]. Other notable SUD PhenomeXcan genes up-regulated in VM astrocytes, and in opioid-exposed VM tissue[10] include the A2B adenosine receptor (*ADORA2B*), which regulates synaptogenesis and synaptic plasticity by downregulating glutamate receptor 5 signaling in astrocytes[45]. Furthermore, the *NFKB2* transcription factor, previously linked to tobacco smoking by transcriptome-wide association, was upregulated in our study in

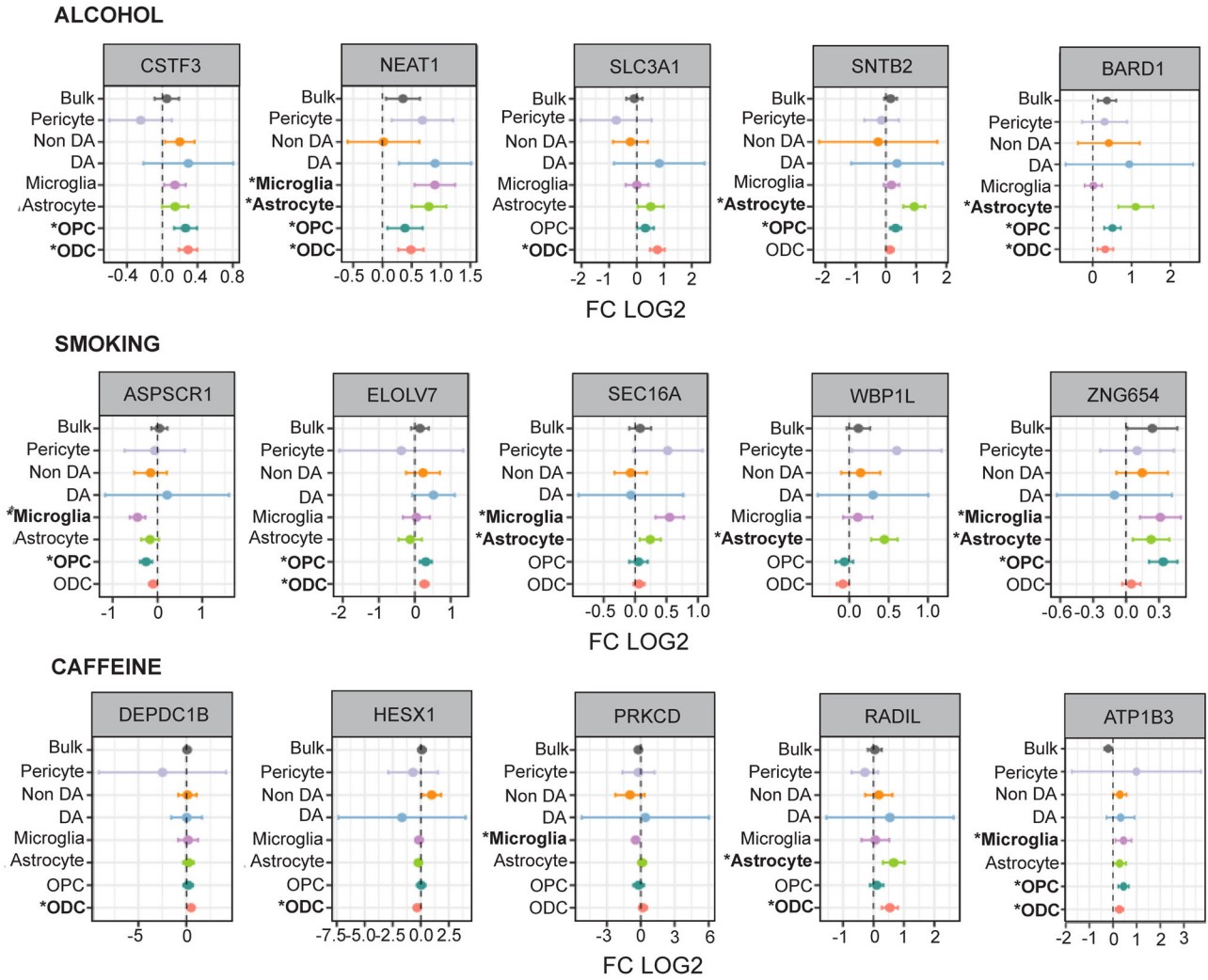

**Fig. 5 | Differentially expressed genes in the VM of opioid users linked to substance use in the general human population.** Forest plots of selected DEGs (FDR *P* < 0.1) in opioid-exposed VM (x axis log fold change compared to control, y-axis VM bulk tissue and single nuclei by cell-type RNA-seq) that are linked to alcohol and caffeine consumption, and smoking in PhenomeX database. Bars represent 95% confidence interval (LFC + /−1.96 S.E.). * bold font marks *P* value from DESeq2 (Wald test following chi-squared distribution with 1 degree of freedom), FDR (Benjamini–Hochberg) *P* < 0.1. See also Data S7. *N* = 95 samples (brain donors). Source data are provided as a Source Data file.

multiple glial subtypes including microglia, astrocytes and OPC of our opioid-exposed VM specimens, again consistent with previous observations made in VM tissue[10]. Interestingly, *NFKB2*, a broad regulator of the genomic response to inflammatory stimuli, is down-regulated in the brains of rats after extinction training for cocaine self-administration[46] in a model for cocaine use disorder. Furthermore, the *Cytokine Inducible SH2 containing Protein (CISH)*, linked to substance use in PhenomeXcan, which was upregulated in microglia of our VM specimens from individuals who died by opioid overdose, and in the VM bulk tissue RNA-seq study[10], encodes a JAK/STAT pathway signaling molecule linked to the anti-inflammatory response in that cell type[47].

In addition to this set of 17 substance use-associated genes that showed significant expression changes in the opioid group, we also identify a set of 312 PhenomeX substance use-associated genes with significant expression changes in multiple glial subtypes of opioid-exposed VM and including several genes linked to the pharmacogenomics of opiate use disorder. For example, *GSG1L* has been linked to plasma methadone levels[48], and the myeloid zinc finger transcription factor MZF1 to cis-regulatory sequences driving expression of the mu-opioid receptor 1[49]. Another noteworthy example of a PhenomeX

substance use-linked candidate gene with altered expression in all three major glial prototypes, including astrocytes, microglia and ODC/OPC (Fig. 5) is the nuclear paraspeckles-associated long non-coding RNA, *NEAT1*, which has been broadly linked to astrocytic and microglial activation[50,51], and has been previous reported as up-regulated in ventral striatum of heroin abusers[52]. Consistent with these findings, multiple inflammatory response- and cell activation- and migration-associated GOs involving *NEAT1*, and additional PhenomeX substance use associated glial DEGs such as *ATP1B3*, *BARD1* and *HESX1* were significantly enriched among the glial DEGs of the present study (Fig. 5, Data S3).

## Discussion

### The glial response to opioid exposure

The present study, which involved individuals with a history of opioid use and dying of opioid overdose matched with drug-free controls, from two independent specimen collection sites in geographically distinct areas of the continental U.S., is one of the largest postmortem studies in this field, profiling the transcriptome in a total of 212,713 single nuclei from the VM. Key findings included up-regulation of pro-inflammatory cytokine and immune response pathways, NFkB

signaling and generalized activation affecting all major glial populations including microglia, astrocytes, and ODCs and their precursors. In contrast, transcript reprogramming in VM neurons, and specifically in the non-dopaminergic (and thus mostly GABAergic) neuronal population, was defined by down-regulated expression of genes involved in synaptic plasticity and neuronal connectivity. Furthermore, there was significant overlap in differentially expressed genes and associated GO pathways between the current VM single nuclei RNA-seq study, and a previous VM bulk (whole-tissue) RNA-seq study[10] of a smaller but distinct cohort of subjects, which showed a similar activation of inflammatory signaling. Thus, the current study identified robust changes in human VM gene expression and associated GO pathways in individuals with a history of opioid use and overdose that are consonant with the previous report but revealed the cell type-specific nature of these changes, which can be characterized as broad glial activation with a prominent representation of immune signaling pathways, in conjunction with downregulation of glial support and neuronal signaling. It is important to emphasize that the aforementioned transcriptomic signatures were independent of the period of specimen collection and geographical location of the cohorts.

Recent transcriptome profiling of bulk tissue from prefrontal cortex (PFC) and nucleus accumbens (using an independent brain collection) reported prominent upregulation of immune signaling genes and transcriptomic signatures indicative of microglial activation and glial motility in the forebrain associated with opioid exposure and overdose[53]. Moreover, a RNA-seq study in ventral striatum of mice exposed to morphine, conducted on FACS-sorted ODCs in conjunction with a general survey of striatal transcription at single nucleus resolution, also broadly resonate with the current findings by demonstrating drug-induced activation of stress and differentiation pathways in all major glial prototypes, with some key genes (e.g., *CDKN1A* and *NFKBIA*) found differentially regulated in the ODC and astrocytes of opioid VM samples in the present study also changed in the opioid-exposed mouse[34] (Data S9). For a subset of differentially regulated glial activation markers, including GFAP, opioid-induced up-regulation in VM was first reported in a rat model three decades ago[54].

These studies, taken together, lead to a consistent consensus implicating glial activation with up-regulation of inflammatory and immune signaling pathways across multiple nodes in the addiction circuitry of the opioid exposed brain, including the VM. Therefore, it is very interesting that drugs acting as inhibitors for the pro-inflammatory glial response reduce morphine-induced withdrawal effects in the rat model[55], and attenuate the addictive features of opioids, including positive reward (for example, 'feeling high') and withdrawal-associated symptoms, in human volunteers diagnosed with OUD[56–58]. The precise pharmaco-molecular and -cellular cascades linking opioid addiction and dependence to glial activation remain to be elucidated. Potential mechanisms could include drug-induced activation of TLR4 and other *Toll-like* receptors which, in turn, activate NFkB to drive transcriptional activation of cytokine and chemokine signaling[59,60]. Furthermore, there is evidence for functional interaction between opioid-receptor and NFkB signaling as discussed in[53]. Given that the transcriptomic alterations in striatum of subjects with cocaine use disorder point to an opposite effect, i.e., decreased neuroinflammation[61], interventions against the pro-inflammatory effects of opioids could provide unique drug class-specific therapeutic opportunities.

### Limitations of the present study
The present study provided new insights into cell type-specific gene expression changes in each of the major glial populations, including OPC, ODC, astrocytes, and microglia, within the opioid-exposed VM. However, our experimental design included pooling of up to 4 VM specimens for 10 K nuclei target recovery for each 10x chromium gel bed assay and, in agreement with previous single nucleus RNA-seq

work on human VM[12–14], less abundant cell types such as dopaminergic and non-dopaminergic VM neurons, as well as pericytes, T-cells, and endothelial and ependymal cells, each comprise only a very small fraction (0.8%–3%) of the total population of VM nuclei in the current study. Multiple disease and control samples lacked the minimum number of nuclei from some of these cell types in order to enter DEG analysis by cell type (set at $N \geq 30$ nuclei per cell type, see Methods), reducing the overall power for many of these rarer VM cell populations, Data S2). Future studies that specifically enrich for cell types such as VM dopaminergic neuron nuclei based on fluorescence activated nuclei sorting[62] will be required to more fully elucidate opioid-induced transcriptional alterations for these relatively rare VM cell types.

All opioid users in the present study died from overdose, a common limitation given that virtually all molecular and cellular studies on the brains of subjects with OUD are conducted either exclusively[53,63,64] or overwhelmingly[32] on individuals who died by overdose. However, the broad congruence of neuroinflammatory signatures in glial populations of (non-overdose) animal models for OUD with the glia-specific transcriptomic alterations reported here, strongly suggest that this type of cell-specific activation of immune signaling genes is driven by exposure to the drug and not limited to opioid users who died by overdose. Furthermore, previous studies of SUD subjects[65] have noted that most changes in DEG examined were observed irrespective of immediate cause of death, or perimortem drug levels, suggesting that such changes may represent core pathophysiological changes associated with SUD.

### Neuron-specific alterations in VM from individuals who died by opioid overdose
The present study identified 69 genes dysregulated in the non-dopaminergic neuronal population in the opioid exposed VM (Data S3). These included neuropsychiatric risk genes such as *AUTS2* and *NCALD*, which reportedly are transcriptionally dysregulated in a VM target tissue, the ventral striatum, after cocaine and amphetamine abuse[66,67]. Furthermore, the aforementioned AP-1 transcription factor *FOS*L2 was up-regulated in VM non-dopaminergic neurons. We counted 9/69 (13%) of differentially regulated neuron-specific genes in VM in our opioid group that matched to neuron-specific enhancer or promoter sequences reportedly affected by histone hypo-acetylation in PFC neurons from individuals who died by opioid overdose[68] (Data S10). These included genes conferring heritable risk for nicotine and other substance dependence, such as *GABBR2* encoding a GABA$_B$ receptor, and *SHC3* involved in MAP kinase and neurotrophin signaling[69,70]. These findings then further support the emerging hypothesis that opioid exposure and addiction is associated with a coordinated transcriptional dysregulation in various neuronal and non-neuronal subpopulations residing in specific nodes of addiction circuitry including VM, striatum, and PFC[71,72].

We predict that future studies, by constructing a neurogenomic atlas of cell specific gene expression alterations and related changes in epigenomic regulation, across multiple regions of opioid-exposed brains and conducting integrative analyses combined with the emerging genetic risk architecture for substance use disorders, will provide deep insights into neuronal and glial mechanisms highly relevant to the neurobiology and treatment of opiate addiction.

## Methods
### Disease and control brains were collected within two separate geographical areas of the U.S., representing the greater Detroit and Miami metropolitan areas. The two collection sites operated independently. Cause of death was determined by forensic pathologists following medico-legal investigations evaluating the circumstances of death including medical records, police reports and scene investigations, autopsy results, and toxicological data. Inclusion in the opioid abuse

group was based on a documented history of opioid abuse, toxicology report positive for opioids, and forensic determination of opioids as the cause of death. Individuals with an identified history of a neurological or psychiatric disorder, a debilitating chronic illness, death by suicide, or evidence of neuropathology at autopsy, were excluded from the study. We note that polydrug use and drug deaths involving combinations of opioids and nonopioid psychoactive drugs are quite common. Forensic evidence, however, supports a history of opioid use for each diseased individual included in the present study. We note that in both the Detroit and Miami cohorts, approximately one-quarter of opioid users were also positive for benzodiazepines and/or other sedatives (a common and particularly deadly drug combination). Concurrent use of opioids and cocaine has recently emerged as a troubling national trend, but none of the Detroit cohort, and only one-fifth of the Miami cohort tested positive for cocaine use within several days of death (as evidenced by detection of cocaine metabolites including benzoylecgonine in urine).

### Detroit cohort

All studies were approved by the Institutional Review Board of Wayne State University. Human midbrain specimens were collected during routine autopsy by the Wayne County Medical Examiner as part of the autopsy process mandated by the laws of the State of Michigan. All disease and control brains were de-identified specimens not requiring consent for the purposes of the present study. Medicolegal investigations were conducted by forensic pathologists. The cause and manner of death were determined after evaluating the circumstances of death, toxicology data, and autopsy results[10,35,73]. Data S1 (columns M, N, O) list toxicology information case-by-case, including blood levels for specific opioids and their metabolites, for ethanol and additional common non-opioid drugs of abuse (e.g., alcohol, cocaine, cannabinoids, anxiolytics, barbiturates). Subjects testing positive for cocaine were excluded from the Detroit cohort of the present study. A subset of our diseased individuals tested positive for opioids plus benzodiazepines, as this reflects a drug class commonly co-abused with opioids with deadly consequences[74]. Individuals in the control group had no documented history of drug abuse, and tested negative for opiates, cocaine, and other drugs of abuse or CNS medications at time of death. Causes of death for control subjects were primarily cardiovascular events or gunshot wounds. Exclusion criteria for either group included a known history of neurological or psychiatric disorder, death by suicide, evidence of neuropathology at autopsy, debilitating chronic illness, estimated postmortem interval (PMI) >20 h, or biochemical evidence of poor tissue sample quality or prolonged perimortem agonal state (i.e., brain pH < 6.2). To reduce variance unrelated to drug abuse, the two groups were matched in terms of sex, race, age, and brain pH at the time of processing. Data S1 and Table S1 include demographic and sample quality characteristics. Brains were sectioned transversely at the level of the posterior edge of the diencephalon and mid pons, to obtain a tissue block encompassing the entire human midbrain[75,76] (corresponding approximately to plates 51–56 of ref.[77]). From this block, the VM region comprised of SN (A9) with adjacent VTA (A10) (Fig. S10A) was processed according to the single nuclei RNA-seq protocol described below.

### Miami cohort

All studies were approved by the Institutional Review Board of Nova Southeastern University, with next of kin consent. Study subjects were selected from an opportunistic sample of opioid intoxication deaths defined by circumstances of death and forensic and supplemental toxicology data. All diseased individuals and unaffected controls were evaluated to rule out comorbid psychopathological diagnoses. Common drugs of abuse and alcohol and positive urine screens were confirmed by quantitative analysis of blood and brain (Data S1, columns M, N, O). Retrospective chart reviews were conducted to confirm

history of opioid abuse, methadone or addiction treatment, drug-related arrests or drug paraphernalia found at the scene. Supplemental brain toxicology was done on select individuals for comparison to blood levels at the time of death. Inclusion in the opioid group was based on a documented history of opioid abuse, toxicology report positive for opioids, and forensic determination of opioids as cause of death. The detection of 6-acetyl morphine (6-AM) was taken as definitive evidence of acute heroin exposure. Drug-free control subjects, with negative urine screens for all common drugs and no history of licit or illicit drug use prior to death, and with no known history of neurological or psychiatric disease, were selected from accidental (motor vehicle accidents or trauma) or cardiac sudden deaths. All diseased and unaffected control individuals were selected from persons who died suddenly without a prolonged agonal state, since agonal state affects brain tissue quality control metrics. Care was taken for cohort selection to match subject groups as closely as possible for non-Hispanic Caucasian ancestry, age, sex, PMI, and brain pH (Data S1, Table S1).

### Sample processing

Sample processing for both brain cohorts, including purification of nuclei, RNA extraction, and generation of single nuclei RNA-seq libraries, was performed in New York. Brains were processed in pools of $N = 3$–4 unique brains (donors). From each unique brain, a tissue aliquot, containing approximately 20 mg of VM from the area of the SN (A9) and portions of the adjacent VTA (A10), was homogenized using a douncer at least 20x in 1 ml lysis buffer (0.32 M sucrose, 5 mM CaCl2, 3 mM Mg(Ace)2, 0.1 mM EDTA, 10 mM Tris pH8, 0.5 mM DTT, 0.1% Triton X-100) with 400U RNase inhibitor (Takara Bio Recombinant RNase Inhibitor, Cat. 2313) added to it. Then, an additional 4 ml of lysis buffer was added and the sample solution was dounced an additional 20x until homogenous. After douncing, each pool of 4 unique samples was transferred to an ultracentrifuge tube (Beckman Colter Polypropylene Centrifuge Tubes $5/8 \times 3\ 3/4$ in., ref. 361707) and underlaid with 9 ml of sucrose buffer (1.8 M sucrose, 3 mM Mg(Ace)2, 0.5 mM DTT, 10 mM Tris-HCl pH8), then ultracentrifuged with 24,000 rpm in a SureSpin 630 (17 mL) Rotor (106,803 x $g$) for 1 h at 4 °C. After centrifugation, the supernatant was removed and each pellet of nuclei was carefully resuspended in 1 ml of 1% BSA with 1000U RRI added to it, transferred to a sterile tube, and 1 μl of the nucleophilic dye, DAPI (4′,6-Diamidino-2-Phenylindole, Dihydrochloride, Invitrogen Cat. D1306), was added. For FACS collection, sterile tubes were coated with 5% BSA. After residual BSA solution at bottom of tubes was removed, DAPI+ nuclei were sorted into the collection tubes using a BD FACSAria Cell Sorter, with approximately 300,000 DAPI+ nuclei for each pool of 4 unique midbrain samples collected and processed using the 10x Chromium Next GEM Single Cell 3′ v3.1 (Dual Index) Protocol (CG000315 Rev A) according to the manufacturer's instructions. The Agilent 2100 High Sensitivity DNA Bioanalyzer Kit was used as a quality control step at Step 2.4 and end of the library preparation, also as per 10x Genomics' guidelines. To prepare samples for sequencing, sample concentration was determined using the KAPA Biosystems Library Quantification Kit (ROX Low qPCR Master Mix, Cat. KK4873). Libraries were sequenced by the New York Genome Center using the Illumina NovaSeq platform aimed at a sequencing depth of 50,000 read pairs per nucleus. Libraries consisted of paired-end reads with a read length of 100 bp.

### Genotyping

Miami Cohort: Genomic (g) DNA was extracted from cerebellar tissue using a QIAamp DNA Micro Kit, followed by SNP genotyping was done with Illumina's MEGA multiethnic array at Rutgers University Cell and DNA Repository (RUCDR Infinite BiologiX).

Detroit Cohort: DNA was extracted from 25 mg of tissue using QIAamp DNA mini kit from Qiagen Cat# 51304 (Qiagen, Germantown,

MD). Tissues were lysed manually and then processed through the QIAcube DNA isolation protocol. For genotyping Infinium Global Diversity Array-8 v1.0 Kit microarrays were processed by the Advanced Genomics Core of University of Michigan (Ann Arbor, MI, USA). Genotype information was converted to vcf format using "iaap-cli gencall" and "gtc_to_vcf.py" from Illumina.

For both cohorts imputation was performed using the TOPMed Imputation Server version 1.5.7 (https://imputation.biodatacatalyst.nhlbi.nih.gov) to a total of 292,140,970 genetic variants. The vcf files from the two cohorts were then merged and filtered for high-quality imputation and coverage for at least ten scRNAseq transcripts using bcftools resulting in a vcf file with 8,184,813 genetic variants.

### scRNA-seq raw data processing (Alignment and demultiplexing)
We processed each scRNAseq library using *cellranger (V7.0) count* using the GRCh38 human reference genome and the default parameters except for the --include-introns option which is recommended for single nuclei preparations. The resulting aligned reads bam files were further processed with *demuxlet*[78] together with the genotype vcf files. Demuxlet assigns the most likely individual for each cell barcode based on the reads overlapping genetic variants. Based on the demuxlet assignments, we removed barcodes that were called doublets, ambiguous, or that could not be assigned to an individual that corresponded to the pool that was used to create the library. Any library that did not contain at least one diseased and one control individual was also excluded. While our quality control pipeline had included a mitochondrial read filter to exclude any nuclei with read rates higher than 20%, the vast majority, or 93.7% of our FANS DAPI sorted single nuclei exhibited a mitochondrial read fraction of <1%, and only 1.8% of nuclei showing a mitochondrial fraction above 2% (Fig. S10B). These filtering procedures resulted in a total of 212,713 high-quality cells with 36,601 genes across 95 individuals. A median value of 2008 cells are detected for each sample, with a median value of 8274 reads per cell and 3070 genes per cell (Fig. S1B–D).

To further assess the quality of our single nuclei transcriptome dataset, we assessed by linear regression how tissue quality indicators such as tissue pH and PMI, as well as demographic variables such as sex and genetic ancestry (summarized by the first two genotype PC), affect the number of nuclei/individual, the number of reads/nucleus, the number of genes/nucleus, and the percentage of mitochondrial genes in each of our single nuclei transcriptomes that had passed all quality controls. Figure S10C shows for tissue pH a significant positive correlation with the number of nuclei per individual, and the number of reads and number of genes in each nucleus ($R = 0.275$–$0.328$) and a negative correlation with the percentage mitochondrial genes ($R = -0.267$). In contrast, PMI and any of the demographic variables showed very weak, and mostly non-significant associations with nuclei number, or reads and genes / nucleus (Fig. S10C)."

### Clustering analysis and cell type annotation
We employed the standard pipeline of *Seurat R package (v4.0)* to further process our scRNA data. After merging all the libraries into one Seurat object, we first utilized the log1pCP10K approach to normalize data and then standardize the gene expression across cells together. For dimensionality reduction analysis, we performed PC analysis on 2000 highly variable genes to obtain 100 PCs. To correct for batch/libraries effects, we calculated the harmony adjusted PCs using *RunHarmony* with the parameter of group.by.vars set to library ID[79]. The UMAP was used to visualize our scRNA data using the top 50 harmony-adjusted PCs. Prior to clustering the cells, we construct a KNN graph based on the euclidean distance using the top 50 harmony-adjusted PCs by running FindNeighbors. Following that, we applied the Louvain algorithm implemented in FindClusters with 0.07 resolution to group cells into 14 distinct clusters.

To annotate cell-type for the scRNA data, we performed DEG analysis between the compared cluster and the remaining clusters (grouped together) to identify cluster-specific expressed genes using FindAllMarkers with default differential test approach Wilcoxon Rank Sum test. For each compared pair, we only focused on the genes with higher expression in the compared one relative to the contrast one (only.pos in FindAllMarkers setting TRUE). To increase more signals involved, we used the relaxed threshold values of at least 2% percent of expressed cells for genes (min.pct) and at least 0.1 fold difference (log-scale) between the two groups of cells (logfc.threshold).

The canonical cell-type marker genes were expected to be highly expressed in the corresponding clusters (Figs. S1E, F, 1B): (1) ODCs marker genes (such as *MOBP, MBP, PLP1* and *CNP*) are highly expressed in clusters 0, 5, 10 and 13; (2). astrocyte-marker genes (such as *GFAP, AQP4* and *SLC1A2*) are mainly enriched in the cluster 1; (3) microglia-marker genes (such as *C3, CSF1R, CX3CR1, LRRK1, DOCK8* and *P2BY12*) are highly expressed the cluster 2 and 11; (4) OPC-marker genes including *VCAN, PDGFRA, OLIG1 and OLIG2* are mostly expressed in the cluster 3; (5) dopaminergic neuron (DaN)-specific genes such as *DDC, SLC6A3, SLC18A2,* and *SLC18A2* tend to be highly expressed in the cluster 7; (6) Non-dopaminergic (Non-DA) neuron genes including *GAD1, GAD2* and *SLC17A7* gene family tend to be highly enriched in the cluster 4; (7) pericytes-marker genes (*MYO1B, NR4A2, PDGFRB* and *RGS5*) are highly expressed in the cluster 6; (8) While endothelial-specific genes such as (*CDH5, CLDN5, FLT1, KDR, PECAM1* and *PTPRB*) are highly expressed in cluster 8; (8) We also identified some T cell specific genes including *CD96, IL7R, SKAP1* and *THEMIS* highly expressed in the cluster 9; (9) For the cluster 12, consisting of very few cells, the genes including *HTR2C* and *TTR* are highly enriched, which are related to the function of ependymal cell.

We also inspected our dataset with an automatically generated cell-type annotation using an independent high quality reference data set[14] collected from human SN. This reference set is comprised of 387,483 cells across 18 samples, annotated by seven major cell types, including ODC, astrocyte, microglia, OPC, dopamine neurons (DA), Non-DA neurons and endothelial cells[14]. We constructed a heatmap to visualize in our dataset the proportion of different cell types accounting for each cluster (Fig. S1G). We note that the above 7 major cell types are dominant in the corresponding clusters that highly expressed cell-type marker genes. After combining the cell type marker genes and automatic cell-type annotation, eventually we assigned the cluster 0, 5, 10 and 13 to be ODC, the cluster 1 as astrocytes, cluster 2 as microglia cells, cluster 3 as OPC, cluster 4 as Non-DA neuron, cluster 6 as pericytes, cluster 7 as DaN, cluster 8 as endothelial cells, cluster 9 as T-cells and cluster 12 as ependymal-cells. This cell-type annotation was used for all downstream analyses.

### Differential gene expression analysis
We generated the pseudo-bulk counts data by summing the reads for each gene across cells that were from the same sample and cell-type. Focusing on the autosomal genes, we obtained a counts matrix of 30,801 genes in 531 combinations of the sample and cell-types with at least 30 nuclei (in pilot studies, we varied the minimally required number of nuclei per cell type and sample from 20 to 100, and determined $N = 30$ nuclei per sample and cell types as minimum number to enter into the disease vs. control DEG analysis because smaller N's increased overall noise factor in the DEG and higher N's were overly restrictive by excluding larger number of subjects for some of the rare cell types). Combinations were also eliminated if the remaining batches did not include at least one individual from both the control and opioid group. This resulted in 7 cell-types considered for the final DEG analysis. For four of these cell-types, including the 4 glial populations that were the focus of the present study, OPC, ODC, astrocytes and microglia, we have a large number of individuals, while for the remaining three cell types, including non-DA and DA neurons

and pericytes, the number of individuals is smaller and the statistical power is more limited (Data S2). We performed DEG analysis between opioid and control groups for each cell-type separately using R *DESeq2 package*[80] and the following model

**Gene expression~opioid+library+sex + pH+age+genotype PC1+genotype PC2+genotype PC3**

which would correct the effects arising from various experimental batch, genetic ancestry and other covariates (sex, age and pH). Note that DESeq2 normalizes the pseudo-bulk input (using default settings) which implicitly will accommodate for differences in the sequencing depth as well as the number of cells that went to each sample/cell-type combination. To filter out lowly expressed genes, we only considered genes that were expressed higher than 0.5 CPM in at least 3 samples from opioid and control (respectively) across libraries. To correct for multiple hypothesis testing, we used the Benjamini–Hochberg approach implemented in *results* function in the DESeq*2 package* with default parameters. We defined the differentially expressed genes as those with FDR < 10% and the fold change at least 1.189 (| log2FC| > 0.25).

### Gene Ontology (GO) enrichment analysis
Using the enrichGO function from *ClusterProfiler (4.0) R package*[81], we performed GO enrichment analysis for DEGs from the 6 cell types for up-regulated and down-regulated genes separately which performs an over-representation test using a one-sided Fisher's exact test, and using all tested genes in DEseq2 as background. To correct for multiple hypothesis testing, we applied the Benjamini–Hochberg approach to calculate the FDR across all the 12 conditions (6 cell types × 2 directions) implemented in *p.adjust* in R (4.1). The significantly enriched biological process terms are defined as those terms (gene size ranging from 5 to 500) with FDR < 10%.

### Transcriptome-wide association analysis (TWAS)
To investigate whether DEGs are associated with the genetic risk variants for substance use traits and disorder (SUD), we screened the *PhenomeXcan database*, a resource for transcriptome-wide association studies linking genes to phenotypes by genetically predicted variation in gene expression[40]. We focused on SN gene expression and population-scale SUD phenotypes in PhenomeXcan. We focused on a total of 40 SUD-related traits from the following categories of traits or diseases: addiction, alcohol, caffeine, marijuana, and smoking (Data S6). We also analyzed a recent study on identification of addiction risk genes that integrated two eQTL cohorts including GTEx and PsychENCODE with addiction risk[42]. For GTEx, the study conducted TWAS analyses using MetaXcan via integration of eQTL from 13 brain regions and identified a total of 351 addiction risk factor genes (FDR < 10%). For PsychENCODE, using frontal and temporal cortex, TWAS analysis using S-PrediXcan identified a total of 410 addiction risk genes with FDR < 10%[42]. We conducted the proportion test using *prop.test* in R (4.1) to examine whether the DEGs overlapping with SUD are enriched in some cell type or some specific trait.

### Reporting summary
Further information on research design is available in the Nature Portfolio Reporting Summary linked to this article.

## Data availability
Annotated single nuclei RNA-seq data generated in this study have been deposited in the Gene Expression Omnibus (GEO) database under accession code GSE240457. The raw data are, per NIH Genomic Data Sharing Policy, available under restricted access in the database for Genotypes and Phenotypes (dbGAP) under accession code phs003260.v1.p1. Source data are provided with this paper.

## Code availability
Original code and scripts used to analyze the data is available on GitHub.https://github.com/piquelab/sc_brains.

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

## Acknowledgements

We thank the Icahn School of Medicine at Mount Sinai's Flow Cytometry Core (Director: Dr. Jordi Ochando) for providing expertize and support on nuclei sorting, the Scientific Computing group at the Icahn School of Medicine at Mount Sinai for computational resources, and personnel at the New York Genome Center for sequencing support, and Behnam Javidfar for general assistance. This work was supported by NIH National Institute of Drug Abuse R01 DA047880.

## Author contributions

Conceived and designed the study: M.J.B., R.P.-R., F.L., D.C.M., S.A. Processed tissue and generated transcriptomic libraries: T.Y.L., A.V., J.L., A.A., H.M.-M., N.P., K.W., M.I. Provided resources: C.J.S. Computational analysis: J.W., R.P.-R., F.L. Wrote the paper: J.W., R.-P.R., F.L., M.J.B., S.A. Co-senior authors F.L., M.J.B., D.C.M., R.P-R. and S.A. were jointly supervising the project.

## Competing interests

The authors declare no competing interests.

## Ethical approval

The Authors are in support of general ethics and inclusion guidelines for researchers in multi-region collaborations involving local researchers so as to promote greater equity in research
