## [Peer Review File · Nature Communications]

Single nucleus transcriptomics of ventral midbrain identifies glial activation associated with chronic opioid use disorderREVIEWER COMMENTS

Reviewer #1 (Remarks to the Author):

The manuscript reports new findings highlighting the role of inflammatory and synaptic signaling in opioid use disorder from postmortem human brain. The authors used single nuclei RNAseq to investigate: 1) cell type populations in human ventral midbrain in control and opioid subjects; and 2) differentially expressed genes between control and opioid subjects across several meta cell type classes (glia vs. neurons). Overall, the findings highlight a potential role for neuroinflammation and synaptic plasticity in ventral midbrain in opioid use disorder and opioid overdose. These findings are consistent with recently published findings using other 'omics approaches from other research groups seemingly. While there are several strengths of the manuscript, there are a number of weaknesses which detract from the overall impact of the approach and findings. These strengths and weaknesses are outlined below:

Strengths

- 1) Ventral midbrain is challenging to reliably dissect from postmortem human brain. Single nuclei RNAseq on human ventral midbrain is a valuable resource for multiple research communities.
- 2) Number of subjects for single nuclei RNAseq is comparatively large in the area of substance use disorders and postmortem brain tissue research (95 subjects across case and control). However, caveats lie in the details of the technical approach and analyses (See below).
- 3) Integrative analysis of DEGs by cell type in opioid cases with TWAS highlights a few interesting findings for potential gene by gene involvement in biological phenomena described in previous human and rodent findings related to opioid actions.

Weaknesses

Major

- 1) Nuclei was extracted from the ventral midbrains of 95 subjects (controls and cases). During extraction, nuclei were pooled then prepared for libraries using the 10X Genomics platform. The median capture rate after QC was 2,000 nuclei per sample (not subject since these were pooled). The number of nuclei sequenced per sample is very low and as the authors noted, of course biased towards preferential cell types within the population. That is, per biological sample, glial subtypes dominated the nuclei preparations compared to neuronal cell types. This is common given the discrepancy in cell numbers that are also brain region dependent. However, the results are unsurprising because the majority, if not all, of the transcriptional signal is from glia, and thus, neuroinflammation.
- 2) Given the pooling, the number of samples considered in the analysis is misrepresented - 95 is more like 20-30 subjects, especially given the low number of nuclei captured and sequenced per "sample".
- 3) Given the bias towards glial subtypes, it's unclear as to whether the DEG analysis was weighted or normalized based on cell number per sample. Presumably, if the DEG analysis considered sample, not subject, as the sample size, and normalized to number of cells, the number of DEGs in glial cells between case and control would be reduced, potentially enhancing some detectable signal in other cell types that would be worth describing. Again, the low number of nuclei per sample may prevent this analysis.

- 4) The lack of shift in cell populations is unsurprising between controls and cases, hindered by the low number of nuclei per sample.
- 5) There is very little discussion as to how these cell clusters align with previous datasets in controls. Believe this is a missed opportunity for aligning with other species where similar, more high resolution, datasets are available (e.g., rat VTA).
- 6) There is surprisingly very little information in the metadata on the cohort used. Deeming an opioid case as "opioid" is not only uninformative, but misses an opportunity to include any other clinical and/or psychiatric diagnosis for future analysis. Most opioid cases in Detroit and Miami banks include polysubstance use, with medical comorbidities that should be considered for not only this manuscript but future analyses (e.g., Meta-analysis). Details regarding poly toxicity at time of death should also be included. Polytoxicity was measured via urine analysis.
- 7) In line with the above, the cohorts are unmatched across controls and cases. PMI, RIN, sex, race, pH are not analyzed across groups, nor are they matched across groups, nor are they considered in the DEG analysis. In addition, sex is remarkably unbalanced here - 84 males to 11 females. Sex is not controlled for in the analysis.

Minor

- 1) More information is needed regarding the cell type annotation transfers. The human SN datasets were used despite this being a ventral midbrain dissection that as the authors note, included SN and VTA. Were only canonical cell type markers for SN included in the analysis? The resolution of the dataset is low.
- 2) Neuroanatomical maps of where the dissections were removed should be included.
- 3) TWAS integration should include newer SUD GWAS datasets - recent publication in Nature by the Agrawal group.
- 4) Limitation of the work is the fact that only subjects with opioid use disorder AND died from opioid overdose were included. Should be addressed in discussion.

Reviewer #2 (Remarks to the Author):

The study by Wei et al investigated cell type-specific gene expression profiles of individuals who died of an opioid overdose compared to controls, and is the largest single cell RNA seq analysis of the ventral mid brain in OUD to date. The study identified potentially important cell-type specific pathways and differentially expressed genes, particularly in glial cells, including pathways that had been previously associated with OUD, and novel genes. Overall, the study is significant in its attempt to identify OUD-associated gene expression alterations in a cell specific manner.

There are some considerations that if addressed would strengthen the study.

1. There are at least 3 cell type clusters not represented by a sufficient proportion of cases and controls (using N=30 as the minimum nuclei # cutoff) to have high statistical power, even though the study included 95 donor samples. There is also a very wide range of number of nuclei per sample, with some samples having as few as 345. Is there a reason why samples with the lowest number of nuclei

representation were not removed?

2. PMI appears to range from 9 to 30 hours and it is not clear whether this was considered as a covariate in analyses. PMI is an important tissue quality covariate that should be adjusted for in differential expression analysis.

3. Analysis should be performed to assess how major human tissue covariates (Age, sex, pH, library, genotype PCs, and PMI) affect nuclei quality indicators: total counts, number of nuclei, mitochondrial gene %.

4. It appears that all non-DA neurons (e.g. excitatory and inhibitory) were pooled together instead of analyzed separately. As OUD has been shown to have different effects on either excitatory or inhibitory neurons, the rationale for this should be explained. It is difficult to remark upon neuronal signaling pathways such as “ionotropic glutamate receptor signaling” in a pool of GABAergic and Glutamatergic neurons.

5. Z-scores for Non-DA neuron correlation between Detroit and Miami sites is very low: $R < 0.1$. This should be commented upon when interpreting the results of DE for non-DA neurons.

6. Please rationalize the use of a 20% mitochondrial gene cutoff for filtering, when 10% is typically standard for quality control. This is especially important, as it appears nuclei were sorted using DAPI FACS, which could yield a lower mitochondrial gene percentage.

7. Please explain whether nuclei were filtered for feature counts as well as mitochondrial gene expression.

8. Please elaborate on how libraries were merged for further processing after demultiplexing.

9. It is not clear what is meant by “for 3 (clusters) the number of individuals is more reduced and the statistical power is more limited”? Which cell types had this problem?

10. It would be useful to comment on the level of expression of opioid receptor genes (MOR, KOR, DOR, NOR), in each cell type.

11. Please elaborate on which clustering algorithm was used, e.g. KNN? The number of clusters (13) seems very small for a clustering resolution of 0.07 in KNN clustering. Also, please explain which differential expression algorithm was used to find cluster-specific DEGs.

Reviewer #3 (Remarks to the Author):

Summary and General Assessment

This study investigates the transcriptional changes in ventral midbrain from human opioid overdose cases in comparison to drug-free control subjects.

While the significance of glia in neural circuit function is increasingly appreciated, glia are still markedly understudied in the addiction field. The present manuscript represents an important, potentially landmark study that demonstrates both the important and potential role of glia in human opioid use disorder.

Overall, this study makes important contributions to our understanding of the glial transcriptome after opioid exposure. However, further analysis can be done with this rich dataset to delineate glial activation and individual cell type differences.

General Comments

- Fig 1 (and Fig S1, oligodendrocyte groups 0, 5, and 13) suggests a big population of oligodendrocytes both in the control and opioid samples. Another study, Marques et al 2016, Science, defines different subgroups of oligodendrocytes in the adult mice brain. Are authors able to detect any subgroups of oligodendrocytes? Could there be differences between control and opioid samples in terms of the DEGs in certain subgroups of oligodendrocytes? Perhaps in terms of metabolic support to axons or myelination pathways? This may provide insights into the role of oligodendrocytes, which might change with opioid exposure.
- Similarly, this study identifies a big population of microglia. Can the authors divide these cells into subgroups in terms of reactive substates states? Comparison to known disease-associated microglial states? This could be potentially very helpful considering they identify an upregulation of inflammatory signatures in multiple cell types.
- How do individual sample brains contribute to forming cell groups in Fig 1 and Fig S1? Can authors see any division between samples within annotated cell groups?
- Fig 1C, and E show that some astrocytes (in green) are grouping with oligodendrocytes (in red) which is different than Fig S1. Could there be a color-coding mistake?
- Throughout the paper, it looks like analysis were done for p adjusted value <0.1 , which causes uninterpretable and confusing results, especially for pathway enrichment analysis. A more stringent statistical threshold is needed. Points below are related to this as well.
- Fig 3 and Fig S3 show Gene Ontology analysis. It would be more helpful if the authors also provided gene ratio for these graphs. Fig 3 shows downregulation of oligodendrocyte and myelin-related genes in astrocytes. Most of the myelin genes are very specifically expressed in oligodendrocytes (such as MBP, PLP1, MOBP etc) therefore it is very confusing to see a downregulation of these myelin-related pathways in astrocytes.
- Fig 3, oligodendrocytes seem to show a downregulation in mitotic cell division. Oligodendrocytes are postmitotic cells, so it is very confusing to see down regulation of these genes. Could authors give more information on the GO analysis, and maybe display only the pathways that are significantly different? (i.e., p-adjusted value <0.05). Because it seems that this downregulation of mitotic division is caused by a few genes (probably less than 5?), and it is not statistically significant (p adjusted not <0.05).

Minor Comments

- Authors used “ODC” as an abbreviation for oligodendrocytes throughout the paper. In glial field, a common short form for oligodendrocytes is “OLs”. Even though it is a minor point, I recommend using “OLs” to keep it consistent with the literature.

- In the introduction section, authors cite a morphine study related to oligodendroglia (Ref 8). This study does not demonstrate the direct effects of morphine in oligodendroglial opioid receptors, but rather shows the role of DA activity-regulated myelin plasticity in morphine-mediated reward behavior. I suggest modifying the last sentence of the first introduction paragraph accordingly.

Point by point response to Reviewer's comments:

Reviewer #1:

The manuscript reports new findings highlighting the role of inflammatory and synaptic signaling in opioid use disorder from postmortem human brain. The authors used single nuclei RNAseq to investigate: 1) cell type populations in human ventral midbrain in control and opioid subjects; and 2) differentially expressed genes between control and opioid subjects across several meta cell type classes (glia vs. neurons). Overall, the findings highlight a potential role for neuroinflammation and synaptic plasticity in ventral midbrain in opioid use disorder and opioid overdose. These findings are consistent with recently published findings using other 'omics approaches from other research groups seemingly. While there are several strengths of the manuscript, there are a number of weaknesses which detract from the overall impact of the approach and findings. These strengths and weaknesses are outlined below:

Strengths

- 1) Ventral midbrain is challenging to reliably dissect from postmortem human brain. Single nuclei RNAseq on human ventral midbrain is a valuable resource for multiple research communities.*
- 2) Number of subjects for single nuclei RNAseq is comparatively large in the area of substance use disorders and postmortem brain tissue research (95 subjects across case and control). However, caveats lie in the details of the technical approach and analyses (See below).*
- 3) Integrative analysis of DEGs by cell type in opioid cases with TWAS highlights a few interesting findings for potential gene by gene involvement in biological phenomena described in previous human and rodent findings related to opioid actions.*

Weaknesses

Major

1.

Nuclei was extracted from the ventral midbrains of 95 subjects (controls and cases). During extraction, nuclei were pooled then prepared for libraries using the 10X Genomics platform. The median capture rate after QC was 2,000 nuclei per sample (not subject since these were pooled). The number of nuclei sequenced per sample is very low and as the authors noted, of course biased towards preferential cell types within the population. That is, per biological sample, glial subtypes dominated the nuclei preparations compared to neuronal cell types. This is common given the discrepancy in cell numbers that are also brain region dependent. However, the results are unsurprising because the majority, if not all, of the transcriptional signal is from glia, and thus, neuroinflammation.

Response:

We appreciate this comment. We note that the Reviewer is confusing 'sample' (we studied N = 95 samples, 1 sample = 1 subject) with the experimental pooling of samples prior to loading the 10x instrument. Because sample identity is preserved and each cell type is assigned to an individual sample (subject) based on the genotype information, the effective sample size after sequencing is unchanged (95 samples or subjects). To minimize any future confusion we now made crystal clear in the title of Table S3, in the legend of Table S1 and in the main text (first chapter of the results section) that we studied N=95 samples, and as shown in Figure S1B and Table S3, that the **median of nuclei number/sample (per subject) is around 2,000 nuclei.**

We would like to remind the Reviewer that our study is to date, to the best of our knowledge, the largest single nuclei RNA-seq study in the substance abuse field, certainly in the context of postmortem brain work.

We write in the first paragraph of the Results section:

"Nuclei were processed in pools of 3-4 brains of cases mixed with controls, using the 10X Chromium system followed by Illumina sequencing, read alignment and processing by 10X Cellranger. Each single nucleus was matched to a donor using Demuxlet, confirming a 100% match by donor by pool against the background of all 95 donors/95 samples (1 sample/donor) (**Figure S1A**). After removal of doublets and quality-control filtering (see Methods), we obtained a total of 212,713 transcriptionally profiled single nuclei, each unique to a singular donor (median, 2,008 nuclei/donor). We collected 2,696~21,363 (median, 8,274) reads/nucleus (**Table S3**) and measured the expression of 1,383~5,079 (median, 3,070) genes/nucleus. Total numbers of single

nuclei/specimen, genes called/single nucleus/specimen and read depth/nucleus/specimen showed no significant differences between cases and controls (**Figure S1B-D**)."

Each VM sample included both substantia nigra and the adjacent ventral tegmental area (SN/VTA) (see Figure S1 and Methods). Nuclei were processed in pools of 3-4 brains of cases mixed with controls, using the 10X Chromium system followed by Illumina sequencing, read alignment and processing by 10X Cellranger."

And we continue:

"We then asked whether opioid exposure altered the proportions of various cell types, including of the various glial populations that were the focus of the present study. In controls, oligodendrocytes (ODC) and their precursors (OPC) taken together comprised 64.3% of all VM nuclei, a proportion that is highly consistent with an independent dataset(14), followed by astrocytes (15.0%) and microglia (13.5%)."

2.

Given the pooling, the number of samples considered in the analysis is misrepresented - 95 is more like 20-30 subjects, especially given the low number of nuclei captured and sequenced per "sample".

Response: Comment 2 of Reviewer #1 is in essence redundant to Comment #1 of the same Reviewer, see our response above. We hope to have clarified any confusion and that in the revised version we have made more clear that we profiled 95 samples from 95 donors (1 sample for each donor), and that this number is unaffected by the pooling strategy used during single cell 10x genomics processing and library preparation.

3.

Given the bias towards glial subtypes, it's unclear as to whether the DEG analysis was weighted or normalized based on cell number per sample. Presumably, if the DEG analysis considered sample, not subject, as the sample size, and normalized to number of cells, the number of DEGs in glial cells between case and control would be reduced, potentially enhancing some detectable signal in other cell types that would be worth describing. Again, the low number of nuclei per sample may prevent this analysis.

Response: Our method implicitly normalizes based on the cell number per sample. This is now made more clear in the methods section. We write "Note that DESeq2 normalizes the pseudo-bulk input (using default settings) which implicitly will accommodate for differences in the sequencing depth as well as the number of cells that went to each sample/cell-type combination."

4.

The lack of shift in cell populations is unsurprising between controls and cases, hindered by the low number of nuclei per sample.

Response: Comment 4 of Reviewer #1 is in essence redundant to Comment #1 and #2. of the same Reviewer, see our response above. Again, we hope we have in the revised version made more clear that we profiled 95 samples from 95 donors (1 sample for each donor).

5.

There is very little discussion as to how these cell clusters align with previous datasets in controls. Believe this is a missed opportunity for aligning with other species where similar, more high resolution, datasets are available (e.g., rat VTA).

Response:

We thank the Reviewer for these important comments. In response, we have conducted additional analysis on neuronal subtypings (presented in new *Figure S2 panels D,E*) and we cite the paper by Philipps et al who conducted very detailed subtyping based on a total of 21,600 nuclei collected from rat ventral midbrain incl. the VTA (incl. at least 399 DA neuron nuclei), to which the Reviewer is apparently referring to. We now write in the Results section, subchapter

"Furthermore, as expected for ventral midbrain, the neuronal subpopulation split into dopaminergic (DA) and non-dopaminergic (Non-DA) (**Figure 1B**), with the former showing

expression for dopamine biosynthetic genes including dopa decarboxylase and tyrosine hydroxylase (*DDC*, *TH*) and the latter separating into a larger subgroup of gabaergic neurons defined by expression of glutamic acid decarboxylase GABA biosynthetic enzymes *GAD1*, *GAD2* and vesicular GABA transporter VGAT *SLC32A1*, and a smaller subgroup of glutamatergic neurons expressing vesicular glutamate transporters *VGLUT1/2* (*SLC17A6/7*). Furthermore, in line with previous studies with single cell resolution in rodent VTA/SN(15, 16), our samples showed considerable heterogeneity for some of the established markers for DA neuron subtyping, including *ALDH1A1*, *SOX6*, and *SLC17A6* (**Figure S2D-E**).

6.

There is surprisingly very little information in the metadata on the cohort used. Deeming an opioid case as "opioid" is not only uninformative, but misses an opportunity to include any other clinical and/or psychiatric diagnosis for future analysis. Most opioid cases in Detroit and Miami banks include polysubstance use, with medical comorbidities that should be considered for not only this manuscript but future analyses (e.g., Meta-analysis). Details regarding poly toxicity at time of death should also be included. Polytoxicity was measured via urine analysis.

Response: We appreciate this comment and in response, have provided additional information to our Table S1, listed in the newly added columns M, N and O. This includes blood levels of (M) specific opioids and their metabolites, (N) alcohol, and (O) other drugs, now also mentioned in the Methods section of the revised paper. We write in the section 'Detroit':

"Table S1 (columns M, N,O) list toxicology information case-by-case, including blood levels for specific opioids and their metabolites, for ethanol and additional common non-opioid drugs of abuse (e.g. alcohol, cocaine, cannabinoids, anxiolytics, barbiturates). Cases testing positive for cocaine were excluded from the Detroit cohort of the present study, while a subset of cases tested positive for opioids plus benzodiazepines, as this reflects a drug class commonly co-abused with opioids with deadly consequences⁵⁷."

and we write in the section 'Miami':

"Common drugs of abuse and alcohol and positive urine screens were confirmed by quantitative analysis of blood and brain (Table S1, columns M, N, O). Retrospective chart reviews were conducted to confirm history of opioid abuse, methadone or addiction treatment, drug-related arrests or drug paraphernalia found at the scene. Supplemental brain toxicology was done on select cases for comparison to blood levels at the time of death. Case inclusion in the opioid group was based on a documented history of opioid abuse, toxicology report positive for opioids, and forensic determination of opioids as cause of death. The detection of 6-acetyl morphine (6-AM) was taken as definitive evidence of acute heroin exposure."

We also added a new paragraph at the beginning of the Methods section. We write:

"Case and control brains were collected within two separate geographical areas of the U.S., representing the greater Detroit and Miami metropolitan areas. The two collection sites operated independently. Cause of death was determined by forensic pathologists following medico-legal investigations evaluating the circumstances of death including medical records, police reports and scene investigations, autopsy results, and toxicological data. Case inclusion in the opioid abuse group was based on a documented history of opioid abuse, toxicology report positive for opioids, and forensic determination of opioids as the cause of death. Cases with an identified history of a neurological or psychiatric disorder, a debilitating chronic illness, death by suicide,

or evidence of neuropathology at autopsy, were excluded from the study. We note that polydrug use and drug deaths involving combinations of opioids and nonopioid psychoactive drugs are quite common. Forensic evidence, however, supports a history of opioid use for each case included in the present study. We note that in both the Detroit and Miami cohorts, approximately one-quarter of opioid cases were also positive for benzodiazepines and/or other sedatives (a common and particularly deadly drug combination). Concurrent use of opioids and cocaine has recently emerged as a troubling national trend, but none of the Detroit cohort, and only one-fifth of the Miami cohort tested positive for cocaine use within several days of death (as evidenced by detection of cocaine metabolites including benzoylecgonine in urine).”

7.

In line with the above, the cohorts are unmatched across controls and cases. PMI, RIN, sex, race, pH are not analyzed across groups, nor are they matched across groups, nor are they considered in the DEG analysis. In addition, sex is remarkably unbalanced here - 84 males to 11 females. Sex is not controlled for in the analysis.

Response: We appreciate this comment. We note that this ratio of male vs. female brains is very typical for human postmortem brain collections, especially those focused on substance use. To address the Reviewer's comment, we now better emphasize that our cohorts are matched and that potential confounds are fully considered in the DEG analyses. We now have added a new table (Table S2) that summarizes, for both the Detroit and the Miami cohorts, the average age in years, male:female ratio, race, and postmortem interval and tissue pH. As documented in the Table S2, in both cohorts, differences between cases and controls were minimal or very modest, and not significant. Furthermore, we write in the Methods section, chapter ‘Differential gene expression analysis’

“We performed differential gene expression analysis between opioid and control groups for each cell-type separately using R **DESeq2 package**⁶² and the following model

Gene expression ~ Opioid + Library + Sex + pH + Age + genotype PC1 + genotype PC2 + genotype PC3

which would correct the effects arising from various experimental batch, ethnicity and other covariates (**sex**, age and pH).”

Minor

1) More information is needed regarding the cell type annotation transfers. The human SN datasets were used despite this being a ventral midbrain dissection that as the authors note, included SN and VTA. Were only canonical cell type markers for SN included in the analysis? The resolution of the dataset is low.

2) Neuroanatomical maps of where the dissections were removed should be included.

3) TWAS integration should include newer SUD GWAS datasets - recent publication in Nature by the Agrawal group.

4) Limitation of the work is the fact that only subjects with opioid use disorder AND died from opioid overdose were included. Should be addressed in discussion.

Response: We appreciate these thoughtful comments and attention to detail by the Reviewer. Regarding minor comment 1) please see our response to Reviewer #1 comment 5 above. Regarding minor comment 2) we now show the region-of-interest in a ventral midbrain map of the new *Figure S10 panel A* and we added the following text and references to the first chapter of the Methods section:

“Brains were sectioned transversely at the level of the posterior edge of the diencephalon to and mid pons, to obtain a tissue block encompassing the entire human midbrain(74, 75) (corresponding approximately to plates 51–56 of DeArmond et al, 1989(77). From this block, the ventral midbrain region comprised of SN (A10) with adjacent VTA (A9) (**Figure S10A**) was processed according to the single nuclei RNA-seq protocol described below.”

Regarding minor comment 3) we now include Agrawal 2023 study in our Results and Methods section and in the Supplemental Figures. In the Methods section, we now have added:

To investigate whether DEGs are associated with the genetic risk variants for substance use traits and disorder (SUD), we screened the *PhenomeXcan database*, a resource for transcriptome-wide association studies linking genes to phenotypes by genetically predicted variation in gene expression(40). We focused on Substantia Nigra (SN) gene expression and population-scale SUD phenotypes in PhenomeXcan. We focused on a total of 40 SUD-related traits from the following categories of traits or diseases: addiction, alcohol, caffeine, marijuana, and smoking (*Table S7*). We also analyzed a recent study on identification of addiction risk genes that integrated two eQTL cohorts including GTEx and PsychENCODE with addiction risk (42). For GTEx, the study conducted TWAS analyses using MetaXcan via integration of eQTL from 13 brain regions and identified a total of 351 addiction risk factor genes (FDR<10%). For PsychENCODE, using frontal and temporal cortex, TWAS analysis using S-PrediXcan identified a total of 410 addiction risk genes with FDR<10% (42). We conducted the proportion test using *prop.test* in R(4.1) to examine whether the DEGs overlapping with SUD are enriched in some cell type or some specific trait. “

In the Results section (subchapter “*Cell-specific DEGs associated with genetic risk for Addiction Disorder*”), we now write:

We focused on brain gene expression and population-scale substance use phenotypes in PhenomeXcan including 40 traits related to caffeine, nicotine, alcohol, and marijuana consumption or dependence(41) (**Table S7**) and, for comparison, a number of medical traits associated with hundreds or thousands of significant genes in the PhenomeXcan resource. We found no specific enrichment of our cell type-specific DEG datasets with traits linked to any specific drug or substance use, or to a general addiction risk score as modeled by Hatoum and colleagues (42) in their recent TWAS studies using GTEx and PsychENCODE datasets as input (**Figure S9A, Tables S8, S9**). However, among the 1,260 PhenomeXcan genes linked to a substance use trait and expressed by at least one cell type in the present study, 325, or 25.8%, were called as significantly altered in our study, including 149 DEGs in microglial nuclei, and 100-70 DEGs in astrocytes and oligodendrocytes and their precursors, respectively, and only very minimal contributions from some of the remaining cell types including pericytes and non-dopaminergic neurons (**Figure S9B Tables S8**).

And Figure S9 has now been revised to include the new data, and the legend of that figure now reads as follows:

Figure S9: DEGs linked to TWAS. (A) Percentages representing, for each cell type the normalized proportion of DEGs overlapping with PhenomeXcan TWAS for substance use and medical or neurological traits, as indicated. We focused on Substantia Nigra (SN) gene expression and population-scale SUD phenotypes in PhenomeXcan. We included a total of 1,260 PhenomeXcan genes for a total of 40 SUD-related traits from the following categories of traits or diseases: alcohol, caffeine, marijuana, and smoking (**Table S7**). We also analyzed a recent study(42) that had called addiction risk genes via TWAS-guided integration of GTEx and PsychENCODE expression quantitative trait loci (see Tables S8, S9). For GTEx, the study conducted TWAS analyses using MetaXcan via integration of eQTL from 13 brain regions and identified a total of 351 addiction risk factor genes (FDR<10%). For PsychENCODE, using the frontal and temporal cortex, TWAS analysis using S-PrediXcan identified a total of 410 addiction risk genes with FDR<10%(42)

Regarding minor comment 4) of Reviewer #1 (Limitation of the work is the fact that only subjects with opioid use disorder AND died from opioid overdose were included. Should be addressed in discussion), very much appreciate this thoughtful comment and in response, have added a new paragraph to the Discussion section of the manuscript, in the subchapter 'Limitations':

"All opioid users in the present study died from overdose, a common limitation given that virtually all molecular and cellular studies on the brains of subjects with opioid use disorder are conducted either exclusively (53, 62, 63) or overwhelmingly(32) on overdose victims. However, the broad congruence of neuroinflammatory signatures in glial populations of (non-overdose) animal models for opioid use disorder with the glia-specific transcriptomic alterations reported here, strongly suggest that this type of cell-specific activation of immune signaling genes is driven by exposure to the drug and not limited to cases with overdose. Furthermore, previous studies of SUD subjects(64) have noted that most changes in differential gene expression examined were observed irrespective of immediate cause of death, or perimortem drug levels, suggesting that such changes may represent core pathophysiological changes associated with SUD."

Reviewer #2:

The study by Wei et al investigated cell type-specific gene expression profiles of individuals who died of an opioid overdose compared to controls, and is the largest single cell RNA seq analysis of the ventral mid brain in OUD to date. The study identified potentially important cell-type specific pathways and differentially expressed genes, particularly in glial cells, including pathways that had been previously associated with OUD, and novel genes. Overall, the study is significant in its attempt to identify OUD-associated gene expression alterations in a cell specific manner.

There are some considerations that if addressed would strengthen the study.

1. *There are at least 3 cell type clusters not represented by a sufficient proportion of cases and controls (using N=30 as the minimum nuclei # cutoff) to have high statistical power, even though the study included 95 donor samples. There is also a very wide range of number of nuclei per sample, with some samples having as few as 345. Is there a reason why samples with the lowest number of nuclei representation were not removed.*

Response: We appreciate this comment and now provide further details in the methods section, subchapter 'Differential gene expression analysis'. As the filter was applied in each cell-type/sample combination separately, we do not think there is any specific reason to further eliminate a sample even if it contributed less nuclei in total, provided that it could contribute at least 30 nuclei for some cell-type. We further ensured that the samples were balanced for cases and controls in each batch.

We write:

"We generated the pseudo-bulk counts data by summing the reads for each gene across cells that were from the same sample and cell-type. Focusing on the autosomal genes, we obtained a counts matrix of 30,801 genes in 531 combinations of the sample and cell-types with at least 30 nuclei (in pilot studies, we varied the minimally required number of nuclei per cell type and sample from 20 to 100, and determined N = 30 nuclei per sample and cell types as minimum number to enter into the case vs. control DEG because smaller N's increased overall noise factor in the DEG and higher N's were overly restrictive by excluding larger number of subjects for some of the rare cell type). Combinations were also eliminated if the remaining batches did not include at least one individual from both the control and opioid group. This resulted in 7 cell-types considered for the final differential gene expression analysis. For four of these cell-types, including the 4 glial populations that were the focus of the present study, OPC, ODC, astrocytes and microglia, we have a large number of individuals, while for the remaining three cell types, including non-DA and DA neurons and pericytes, the number of individuals is smaller and the statistical power is more limited (**Table S3**)."

“Note that DESeq2 normalizes the pseudo-bulk input (using default settings) which implicitly will accommodate for differences in the sequencing depth as well as the number of cells that went to each sample/cell-type combination.”

2.

PMI appears to range from 9 to 30 hours and it is not clear whether this was considered as a covariate in analyses. PMI is an important tissue quality covariate that should be adjusted for in differential expression analysis.

3.

Analysis should be performed to assess how major human tissue covariates (Age, sex, pH, library, genotype PCs, and PMI) affect nuclei quality indicators: total counts, number of nuclei, mitochondrial gene %.

Response: (to 2., 3.) We appreciate this comment. We would like to point out that we have included tissue pH as covariate in our analysis (See Methods chapter ‘Differential Gene Expression Analysis’), because tissue pH is an objective measure for tissue quality, reflective of perimortem brain ischemia and agonal state (PMID17045977; PMID19473297)

We write:

“We performed differential gene expression analysis between opioid and control groups for each cell-type separately using R **DESeq2 package** and the following model ‘Gene expression~Opioid+Library+Sex+pH+Age+genotype PC1+genotype PC2+genotype PC3’”

Please note that postmortem intervals (PMI) for specimens provided by the Medical Examiner’s office are estimates only. Therefore, the Detroit brain collection has, for many years now, taken a pragmatic approach by dichotomizing their brains into two categories, (i) PMI <20 hours and (ii) PMI > 20 hours. As shown in our Table S1, all our Detroit cases and controls are in the former category, with PMI < 20 hours.

In response to this and also the following comment made by the Reviewer, we now have conducted additional analyses. We assessed by linear regression how tissue confounds such as tissue pH and PMI, as well as demographic variables such as sex and genetic ancestry, affect the number of nuclei/individual, the number of reads/nucleus, the number of genes/nucleus, and the percentage of mitochondrial genes in each of our single nuclei transcriptomes that had passed all quality controls. These additional analyses, presented in *Figure S10C*, show that tissue pH shows significant positive correlations with the number of nuclei per individual, and the number of reads and number of genes in each nucleus ($R= 0.275-0.328$) and a negative correlation with the percentage of mitochondrial genes ($R= -0.267$). In contrast, PMI and any of the demographic variables showed very weak, and mostly non-significant associations with nuclei number, or reads and genes / nucleus. Finally, *Figure S10C* shows that the vast number of our single nuclei showed a mitochondrial read fraction of <1%, with no occurrences above 2.5%. We now describe these additional analyses in the Methods subchapter ‘*scRNA-seq raw data processing (Alignment and demultiplexing)*’:

“While our quality control pipeline had included a mitochondrial read filter to exclude any nuclei with read rates higher than 20%, vast majority, or 93.7% of our FANS DAPI sorted single nuclei exhibited a mitochondrial read fraction of <1%, and only 1.8% of nuclei showing a mitochondrial fraction above 2% (**Figure S10B**). These filtering procedures resulted in a total of 212,713 high-quality cells with 36,601 genes across 95 individuals. A median value of 2,008 cells are detected for each sample, with a median value of 8,274 reads per cell and 3,070 genes per cell (**Figure S1B-D**).

To further assess the quality of our single nuclei transcriptome dataset, we assessed by linear regression how tissue quality indicators such as tissue pH and PMI, as well as demographic variables such as sex and genetic ancestry (summarized by the first two genotype principal

components), affect the number of nuclei/individual, the number of reads/nucleus, the number of genes/nucleus, and the percentage of mitochondrial genes in each of our single nuclei transcriptomes that had passed all quality controls. **Figure S10C** shows for tissue pH a significant positive correlation with the number of nuclei per individual, and the number of reads and number of genes in each nucleus ($R= 0.275-0.328$) and a negative correlation with the percentage mitochondrial genes ($R= -0.267$). In contrast, PMI and any of the demographic variables showed very weak, and mostly non-significant associations with nuclei number, or reads and genes / nucleus (**Figure S10C**)."

4. *It appears that all non-DA neurons (e.g. excitatory and inhibitory) were pooled together instead of analyzed separately. As OUD has been shown to have different effects on either excitatory or inhibitory neurons, the rationale for this should be explained. It is difficult to remark upon neuronal signaling pathways such as ionotropic glutamate receptor signaling in a pool of GABAergic and Glutamatergic neurons.*

Response: We agree with the reviewer that ideally we should not mix different types of neurons together, yet as this was a relatively small cluster we could not further dissect them in this study. This should be the focus of a future study that can target a higher number of these types of neurons. In this study we focused mainly on non neuronal cell types.

5. *Z-scores for Non-DA neuron correlation between Detroit and Miami sites is very low: $R < 0.1$. This should be commented upon when interpreting the results of DE for non-DA neurons.*

Response: This cluster is the one for which we have less individuals and the lower correlation value is probably a consequence of this. Note that these correlations are obtained using all genes standardized effect sizes (Z-scores) and include many genes for which the opioid effect may be negligible. However we still find a significant positive correlation even for the Non-DA neurons which is the smallest cluster for which we could do the DEG analysis.

6.

Please rationalize the use of a 20% mitochondrial gene cutoff for filtering, when 10% is typically standard for quality control. This is especially important, as it appears nuclei were sorted using DAPI FANS, which could yield a lower mitochondrial gene percentage.

7. *Please explain whether nuclei were filtered for feature counts as well as mitochondrial gene expression*

Response: (to 6., 7.) We now added a new figure panel (**Figure S10B**) and clarify in the methods section as specified in detail above under response to Reviewer #2 comments 2., 3.

8. *Please elaborate on how libraries were merged for further processing after demultiplexing.*

Response: Libraries were not merged after demultiplexing. Cells were assigned to each sample (subject) of origin based on genotype information. For the differential gene expression analysis we generated the pseudo-bulk counts data by summing the reads for each gene across cells that were from the same sample (subject) and cell-type. This procedure is included in the methods section (first paragraph of subchapter 'Differential gene expression analysis')

9. *It is not clear what is meant by for 3 (clusters) the number of individuals is more reduced, and the statistical power is more limited. Which cell types had this problem?*

Response: We appreciate this comment and now have added in the Methods subchapter, 'Differential gene expression analysis' the following text:

Focusing on the autosomal genes, we obtained a counts matrix of 30,801 genes in 531 combinations of the sample and cell-types with at least 30 nuclei (in pilot studies, we varied the minimally required number of nuclei per cell type and sample from 20 to 100, and determined $N = 30$ nuclei per sample and cell types as minimum number to enter into the case vs. control DEG because smaller N 's increased overall noise factor in the DEG and higher N 's were overly

restrictive by excluding larger number of subjects for some of the rare cell type). Combinations were also eliminated if the remaining batches did not include at least one individual from both the control and opioid group. This resulted in 7 cell-types considered for the final differential gene expression analysis. For four of these cell-types, including the 4 glial populations that were the focus of the present study, OPC, ODC, astrocytes and microglia, we have a large number of individuals, while for the remaining three cell types, including non-DA and DA neurons and pericytes, the number of individuals is smaller and the statistical power is more limited (**Table S3**).

10. *It would be useful to comment on the level of expression of opioid receptor genes (MOR, KOR, DOR, NOR), in each cell type.*

Response: We very much appreciate this constructive comment. We now report in the Results section, subchapter 'Hundreds of glial transcripts show altered expression in opioid-exposed midbrain' as follows:

"Of note, transcripts for each of the four G-protein coupled opioid and opioid-related receptors, including the 'classical 3' OPRM1 (mu) and OPRD1 (delta) and OPRK1 (kappa), plus OPRL1 (nociceptin), were readily detectable among the various glial cell types in the ventral midbrain, with particularly robust expression of OPRM1 in the microglia (**Figure S3**). These findings, which are consistent with previous reports on cell-specific expression, and functional, ligand-binding and mutant-mice studies(17, 18) would suggest that opioid exposure could have direct effects on VM glia in addition to adaptations mediated by drug-related neuronal signaling changes."

11. *Please elaborate on which clustering algorithm was used, e.g. KNN? The number of clusters (13) seems very small for a clustering resolution of 0.07 in KNN clustering. Also, please explain which differential expression algorithm was used to find cluster-specific DEGs.*

Response: We appreciate these comments. We provide detailed information in the Methods section of our manuscript, subchapter 'Clustering analysis and cell type annotation'. Furthermore, we specify in subchapter 'Differential Gene Expression Analysis, that we are using the DESeq2 package after aggregating the reads per cell-type and individual (subject).

Reviewer #3:

Summary and General Assessment

This study investigates the transcriptional changes in ventral midbrain from human opioid overdose cases in comparison to drug-free control subjects.

While the the significance of glia in neural circuit function is increasingly appreciated, glia are still markedly understudied in the addiction field. The present manuscript represents an important, potentially landmark study that demonstrates both the important and potential role of glia in human opioid use disorder.

Overall, this study makes important contributions to our understanding of the glial transcriptome after opioid exposure. However, further analysis can be done with this rich dataset to delineate glial activation and individual cell type differences.

General Comments

1.

Fig 1 (and Fig S1, oligodendrocyte groups 0, 5, and 13) suggests a big population of oligodendrocytes both in the control and opioid samples. Another study, Marques et al 2016, Science, defines different subgroups of oligodendrocytes in the adult mice brain. Are authors able to detect any subgroups of oligodendrocytes? Could there be differences between control and opioid samples in terms of the DEGs in certain subgroups of

oligodendrocytes? Perhaps in terms of metabolic support to axons or myelination pathways? This may provide insights into the role of oligodendrocytes, which might change with opioid exposure.

2.

Similarly, this study identifies a big population of microglia. Can the authors divide these cells into subgroups in terms of reactive substrates states? Comparison to known disease-associated microglial states? This could be potentially very helpful considering they identify an upregulation of inflammatory signatures in multiple cell types.

Response (to 1., 2.): We thank the Reviewer for these excellent suggestions. We conducted unbiased subtyping of the oligodendrocyte, and separately, microglial nuclei populations and then conducted marker gene and differential gene expression for the two major subtypes that emerged for the oligodendrocyte population and the two major subtypes that emerged for the microglia. Results are presented in the new Figure S5A-D and Table S5A,B. Basically, the vast majority of our oligodendrocyte nuclei population separated into one subpopulation expressing marker genes for immature and myelin-forming oligodendrocytes (in good agreement with Marques et al. 2016, the paper referred to by the Reviewer), and a second subpopulation defined by high expression of marker genes that are associated with mature and aged oligodendrocytes. We now write in the Results section, subchapter 'Hundreds of glial transcripts show altered expression in opioid-exposed midbrain', as follows:

“Unbiased subclustering of glial cell types revealed at least two subpopulations of oligodendrocyte, and of microglia nuclei, each of which were broadly represented in the majority of the cases and controls (Figure S5A, Table S5A, B). For example, ODC subcluster '0' was defined by elevated expression of *OPALIN* and other marker genes that define myelin-forming oligodendrocytes, while ODC subcluster '1' showed much higher expression of *S100B*, *RBFOX1* and various other marker genes previously linked to mature (aged) and stressed ODC(12, 19) (Figure S5A,B, Table S5A). However, correlation matrices summarizing differential gene expression between cases and controls across all glial and neuronal subpopulations (Figure S5C), in addition to subtype specific DEG analysis and proportional counts of nuclei in diseased and control brains (Figure S5D), confirm that oligodendrocyte-specific transcriptional alterations are not limited to the subpopulation of aged and stressed nuclei while also affecting younger, myelin-forming ODC. Likewise, correlational analyses confirmed that microglia-specific alterations in overdose cases affect multiple types of microglia (Figure S5C). This included subtype '0', or a large group of microglial nuclei defined by higher expression of interleukin *IL18* and heat shock protein *HSPB1*, two molecules that reportedly promote neuroinflammation in adult human brain(20-22), and higher expression of additional regulators of cytokine signaling such as *SOCS6* which is thought to affect the interaction between microglia and midbrain dopaminergic neurons(23) (Table S5B, Figure S5A).”

3.

How do individual sample brains contribute to forming cell groups in Fig 1 and Fig S1? Can authors see any division between samples within annotated cell groups?

Response: We appreciate this comment and would like to point out that we show in Figure 1E four UMAP plots in a 2 x 2 design defined by diagnosis and brain collection site. As shown in Figure 1E, these four UMAP plots, generated independently, show a strikingly similar profile across all 2 x 2 comparisons, and therefore we feel, Figure 1E and Table S2 (numbers by cell type for each subject) already address the Reviewer's question, by essentially ruling out substantial variation of cell group clustering by diagnosis and collection site.

4.

Fig 1C, and E show that some astrocytes (in green) are grouping with oligodendrocytes (in red) which is different than Fig S1. Could there be a color-coding mistake?

Response: We appreciate this attention to important detail by the Reviewer. We reviewed our methods and we do not believe we made any color-coding mistake. Cluster analysis determines color coding and UMAP the x,y coordinates but they are essentially two different algorithms. Importantly, distances in the UMAP plot

should not be overinterpreted because it uses a nonlinear mapping between the high dimensional gene expression space to just two dimensions akin to a projection that can greatly distort distances in different locations of the plot. The cluster analysis on the other hand uses all the high dimensional space and it is what we use to determine cell-type, color the cells and group them for the subsequent analyses. The UMAP provides a nice visualization of the cells and shows that cells of the same cluster or closely related cell types tend to be close to each other, but again we need to exercise caution in overinterpreting it.

To examine this, we redrew our UMAP plot specifically for the astrocyte and oligodendrocyte population of nuclei (the newly added *Figure S2A-C*). In that UMAP plot, only a minimal number of nuclei at the periphery of the oligodendrocyte cluster were marked as 'astrocytes' (green dots in 'red' ODC cluster).

5.
Throughout the paper, it looks like analysis were done for p adjusted value <0.1, which causes uninterpretable and confusing results, especially for pathway enrichment analysis. A more stringent statistical threshold is needed. Points below are related to this as well.

Response: Using an adjusted p-value of 0.1 corresponding to an FDR of 10% is quite common in genomics. Note that for many of the figures and discussed results including pathway enrichment analysis actually have a lower adjusted p-values than 0.05 so results would remain unchanged. The supplementary tables report all the adjusted p-values as this can facilitate future integrative analyses by others that require all the summary statistics.

6.
Fig 3 and Fig S3 show Gene Ontology analysis. It would be more helpful if the authors also provided gene ratio for these graphs. Fig 3 shows downregulation of oligodendrocyte and myelin-related genes in astrocytes. Most of the myelin genes are very specifically expressed in oligodendrocytes (such as MBP, PLP1, MOBP etc) therefore it is very confusing to see a downregulation of these myelin-related pathways in astrocytes.

Response: We appreciate this comment. We now mention in the legend of Figures 3, S6 and S7 that gene ratios are provided in the accompanying table S6 (column E)

7.
Fig 3, oligodendrocytes seem to show a downregulation in mitotic cell division. Oligodendrocytes are postmitotic cells, so it is very confusing to see down regulation of these genes. Could authors give more information on the GO analysis, and maybe display only the pathways that are significantly different? (i.e., p-adjusted value <0.05). Because it seems that this downregulation of mitotic division is caused by a few genes (probably less than 5?), and it is not statistically significant (p adjusted not <0.05).

Response: We appreciate this very thoughtful comment by the Reviewer. Indeed, as listed in Table S4 and shown in Figure 3 (bottom row), oligodendrocytes from opioid cases show FDR adjusted significant ($P < 0.05$) enrichment of several GO pathways linked to the spindle apparatus and mitotic complex (GO 0000280/67 nuclear division/mitotic nuclear division, and various related pathways such as GO 0007052 (mitotic spindle organization) with FDR adjusted $P < 0.1$). We would like to prefer this more comprehensive version of Table S4, which lists the FDR adjusted P value for each GO category (with cut-off of $P < 0.1$), enabling the reader to assess both the GOs with formal significance ($P < 0.05$) and the related GOs with a statistical trend ($P < 0.1$).

Furthermore, and importantly, we would like to point out that many genes with a role in the mitotic spindle apparatus, including regulators/components of the centriole and related microtubular structures are continued to be expressed in interphase nuclei given their role in a variety of nuclear and cytoplasmic structures and various cellular processes (reviewed, for example, in *Nigg and Raff, Cell 139:663-678 (2009)*). In short, 'mitotic' genes' differentially expressed in oligodendrocytes from our cases could indeed reflect alterations in cytoskeletal regulation and other functions in postmitotic cells. Therefore, based on the Reviewer's comment, we now add the following sentences to the Results section:

“Furthermore, down-regulated expression in ODC from opioid cases included multiple GO-defined mitotic spindle genes (**Figure 3, Table S6**), such as *AURKA*, *FIGNL1*, and *KIF11*. Importantly, the function of these genes extends beyond mitosis as they maintain expression in interphase nuclei to regulate microtubular structures(26-28) and, in case of

TMEM67, are linked to white matter tract alterations in the human midbrain(29). Therefore, altered expression of these genes could indicate potential cytoskeletal alterations in differentiated, postmitotic ODC from overdose cases.”

8.

Minor Comments

Authors used ODC as an abbreviation for oligodendrocytes throughout the paper. In the glial field, a common short form for oligodendrocytes is ???OLs???. Even though it is a minor point, I recommend using OLs to keep it consistent with the literature.

Response: We appreciate this comment but would like to point out that other papers published in Nature journals used ODC as abbreviation for oligodendrocytes (<https://www.nature.com/articles/s41467-020-17876-0>). Therefore, we will keep for now ‘ODC’ as an abbreviation; however if the Editor advises so, or the Reviewer continues to advise to change to ‘OL’, we will implement the change to ‘OL’.

9.

In the introduction section, authors cite a morphine study related to oligodendroglia (Ref 8). This study does not demonstrate the direct effects of morphine in oligodendroglial opioid receptors, but rather shows the role of DA activity-regulated myelin plasticity in morphine-mediated reward behavior. I suggest modifying the last sentence of the first introduction paragraph accordingly.

Response: We appreciate the Reviewer’s comment and have modified the last sentence in the first introduction paragraph accordingly.

REVIEWERS' COMMENTS

Reviewer #1 (Remarks to the Author):

Thank you for responding so thoroughly to each of the previous comments.

Reviewer #2 (Remarks to the Author):

All concerns have been addressed.

Reviewer #3 (Remarks to the Author):

The authors have addressed my comments and have improved the manuscript. It is an important contribution to the literature.

Reviewer #1 (Remarks to the Author):

Thank you for responding so thoroughly to each of the previous comments.

Our Response: Thank you!

Reviewer #2 (Remarks to the Author):

All concerns have been addressed.

Our Response: Thank you!

Reviewer #3 (Remarks to the Author):

The authors have addressed my comments and have improved the manuscript. It is an important contribution to the literature.

Our Response: Thank you!